# Generalized Prediction Set with Bandit Feedback

**Zhou Wang**                                                                    *zwang198@binghamton.edu*
*Department of Mathematics and Statistics*
*Binghamton University, the State University of New York*

**Xingye Qiao**                                                                   *xqiao@binghamton.edu*
*Department of Mathematics and Statistics*
*Binghamton University, the State University of New York*

**Reviewed on OpenReview:** *https://openreview.net/forum?id=VlwqIz41Hp*

## Abstract

In high-stakes environments where uncertainties abound, set-valued prediction offers a cautious and robust mechanism by presenting multiple potential labels as the prediction for each test instance to mitigate the potential risk associated with prediction errors. Yet, integrating this paradigm with out-of-distribution (OOD) detection remains scarcely explored in such settings as online learning with bandit feedback. The bandit feedback mechanism informs the learner about the correctness of the pulled arm/action instead of the explicit ground truth label, leaving the true class label unknown when an incorrect action is taken. To address this challenge, we introduce BanditGPS which conducts set-valued prediction with OOD detection in the bandit feedback setting, using an estimation to the ground truth of class labels. BanditGPS achieves three objectives: render small/informative prediction sets, enhance the OOD detection performance, and control the recall for all normal classes to meet prescribed requirements. Our approach is characterized by the loss function, which trades off between high OOD detection and small prediction sets. Theoretically, we prove that the convergence rate of the regret is $\tilde{\mathcal{O}}(T^{-1/2})$. The empirical results further show that BanditGPS effectively controls the recalls with promising performances on OOD detection and informative prediction.

## 1 Introduction

Conventional single-valued prediction assigns only a single class label to each instance without a provable confidence guarantee. This paradigm could be overly-confident for some data instances, and thereby lead to erroneous decisions with severe consequences. Such an issue is particularly pronounced in critical domains where instances are characterized by high uncertainties. For example, in medical triage, based on a single-label prediction, an AI system might incorrectly categorize a patient with subtle symptoms of a severe condition as low-risk, delaying the critical treatment and hence endangering the patient's life. Similarly, in financial fraud detection, a single-valued prediction system might not adequately account for the nuanced behaviors of complex fraud schemes, resulting in either over-blocking of legitimate transactions (causing inconvenience and potential loss of business) or under-detecting sophisticated fraud tactics (leading to financial losses). To reduce these associated risks from those observations with high uncertainty, set-valued classification, which reports multiple possible class labels, may be employed. This approach allows for human intervention in those difficult data instances, enabling further error reduction through follow-up investigations.

The literature on set-valued prediction is diverse, with several different types of methodologies. (1) Classification with Reject Option (CRO) (Herbei & Wegkamp, 2006; Bartlett & Wegkamp, 2008; Charoenphakdee et al., 2021) incorporates a rejector into the loss function, which is used to abstain from classification if an instance is highly ambiguous. This is equivalent to assigning all labels to the instance, resulting in a less informative prediction set. Zhang et al. (2018) introduces a refined option to produce a more informative/smaller

prediction set. However, CRO and its variants, despite considering the uncertainty of instances, do not control the "accuracy" under the set-valued prediction paradigm, i.e., the probability of the true label being included in the prediction set. (2) In contrast, Conformal Prediction (CP) (Shafer & Vovk, 2008; Vovk et al., 2005) offers a prediction set with a theoretical guarantee of accuracy control. CP, assuming data exchangeability, is a distribution-free framework that works alongside machine learning models, e.g., neural networks, to facilitate set-valued prediction. The caveat with CP is that the prediction set may be less informative due to not explicitly minimizing the set size. Romano et al. (2020); Angelopoulos et al. (2021) have focused on developing score functions to reduce the prediction set size to alleviate this limitation. (3) Confidence Set Learning (CSL) (Wang & Qiao, 2018; 2022a; Sadinle et al., 2019) approaches set-valued prediction from a constrained optimization perspective, explicitly minimizing the prediction set size while controlling accuracy. Conversely, Denis & Hebiri (2017; 2020) seek to maximize the accuracy of the prediction given a size budget.

All the above three camps concentrate on set-valued prediction within a closed-world assumption, where each instance is guaranteed at least one known class label. Nevertheless, the dynamic nature of the open world presents the inevitability of encountering new, unknown classes over time. This reality necessitates the unified process of not only classifying known entities but also detecting novel classes, a task termed out-of-distribution (OOD) detection or open-set recognition (Yang et al., 2021; Lee et al., 2018). Traditional approaches to OOD detection encounter limitations akin to those of single-valued prediction systems. In an effort to bridge this gap, by admitting an empty set for potential atypical points, recent innovations have expanded set-valued classification to incorporate an OOD detection component, such as Cautious Deep Learning (CDL) (Hechtlinger et al., 2018), Balanced Conformal Prediction Set (BCOPS) Guan & Tibshirani (2022), Generalized Prediction Set (GPS) (Wang & Qiao, 2023; 2022b), and Deep Generalized Prediction Set (DeepGPS) (Wang & Qiao, 2024a). CDL and BCOPS, rooted in Conformal Prediction (CP), unfortunately inherit its predisposition towards larger prediction sets. Conversely, GPS and DeepGPS strive for informative prediction by explicitly minimizing prediction set sizes, aligning with the principles of Confidence Set Learning (CSL).

However, these advancements predominantly address scenarios of offline learning with full feedback, where detailed label information for each instance is accessible—even following an incorrect prediction. This starkly contrasts with the online bandit feedback setting, where a learner's received feedback is limited to the binary outcome of an arm/action's success, devoid of explicit insights across all labels. For example, within clinical trials, an adaptive trial design system (learner) might select a treatment (arm), e.g., a type of drug, without knowing its efficacy beyond the absence of disease remission, thus lacking comprehensive feedback on optimal treatments. Although Bandit Class-specific Conformal Prediction (BCCP) (Wang & Qiao, 2024b) represents a step towards accommodating set-valued classification in a closed world under bandit feedback, it lacks mechanisms for OOD detection. Given the limited research in this domain, our study pioneers the exploration of generalizing the set-valued prediction to have the capacity of OOD detection under the bandit feedback setting.

In this article, we have made the following four major contributions. Methodologically, we introduce BanditGPS, an adaptation of the Generalized Prediction Set for online learning with bandit feedback, marking a novel intersection of these research areas. Without accessing the explicit label information, the proposed approach handles the bandit feedback by utilizing an estimation of the ground truth of the class label. Secondly, unlike the offline training regime of DeepGPS (Wang & Qiao, 2024a), BanditGPS dynamically adjusts to maintain class-specific recall (to be defined later) for normal classes with fewer manual parameter adjustments, leading to a feasible deployment for real-world online learning applications. Thirdly, unlike its predecessors, the BanditGPS allows explicit control of the balance between informative prediction and OOD detection effectiveness, making it easy for practitioners to use given their specific business needs. Lastly, we theoretically prove a regret convergence rate of $\tilde{\mathcal{O}}(T^{-1/2})$ which is competitive in the literature; the empirical study confirms the efficacy of BanditGPS.

The rest of this article is organized as follows. In Section 2, we delve into key notations and review works that lay the groundwork for our study. Section 3 articulates the problem formulation and introduces the BanditGPS method with an algorithm. The convergence rate of regret for the algorithm is theoretically explored in Section 4. In Section 5, we present empirical evidence demonstrating the effectiveness of BanditGPS. A

conclusion is offered in Section 7. Additional materials, including detailed proofs and extended experiments, are provided in Section A.

## 2 Preliminary

In this section, we discuss some notions and briefly review seminal works across three pivotal areas: set-valued classification, out-of-distribution detection, and the multi-armed Bandit problem.

**Set-valued Classification:** Let $(\boldsymbol{X}, Y) \in \mathcal{X} \times \mathcal{Y}$ be an observation generated from an unknown distribution, where $\mathcal{Y} = \{1, \ldots, K\}$. A set-valued classifier is a map $\phi : \mathcal{X} \mapsto 2^{\mathcal{Y}}$ to return a prediction set $\phi(\boldsymbol{X})$ for the instance $\boldsymbol{X}$ based on a certain rule. Commons metrics (Vovk et al., 2017) to evaluate the performance of a set-valued classifier include the (expected) prediction set size and the accuracy. The prediction set size is defined through the cardinality of the prediction set, i.e., $|\phi(\boldsymbol{X})|$, which gauges the number of class labels predicted for instance $\boldsymbol{X}$. Because both the classifier $\phi(\cdot)$ and the instance $\boldsymbol{X}$ are random, we typically focus on $\mathbb{E}[|\phi(\boldsymbol{X})|]$. The accuracy in the set-valued prediction paradigm is defined as the probability that the true label of an observation is included in its prediction set, i.e., $\mathbb{P}[Y \in \phi(\boldsymbol{X})]$, which is also referred to as coverage probability in the CP literature (Vovk et al., 2005).

The accuracy $\mathbb{P}[Y \in \phi(\boldsymbol{X})]$ represents the overall performance of the classifier over all classes. For each class, we denote the (class-specific) recall as $\mathbb{P}[Y \in \phi(\boldsymbol{X}) \mid Y = k]$ to measure the classifier's performance on class $k$. Intuitively speaking, there is a trade-off between prediction set size and recall. Higher recall for normal classes often comes with a larger prediction set $\phi(\boldsymbol{X})$ (because this allows an increased likelihood for the true class label being included in $\phi(\boldsymbol{X})$). While CP-based classifiers (Vovk et al., 2005; Romano et al., 2020; Angelopoulos et al., 2021) offer a controlled recall guarantee, their prediction set sizes are contingent on the chosen score function. In contrast, CSL-based approaches (Sadinle et al., 2019; Wang & Qiao, 2018; 2022a; 2023) strive to minimize prediction set sizes while maintaining the normal class recall.

**Out-of-distribution (OOD) Detection and Open-set Recognition (OSR):** The goal of OOD and OSR (Bendale & Boult, 2016; Lee et al., 2018; Charpentier et al., 2020; Yang et al., 2021) is to conduct single-valued prediction for instances in the normal classes in addition to rejecting atypical observations that potentially comes from a new or anomaly class that was not present in the training data. Recently, Selective Classification with OOD detection (SCOD) (Xia & Bouganis, 2022; Zhu et al., 2023) and Unified Open-set Recognition (UOSR) (Kim et al., 2023; Cen et al., 2023) further consider identifying difficulty observations (hard to distinguish among existing normal classes) when they are not rejected as OOD points. The difficult observations and OOD points in both SCOD and UOSR can be respectively viewed as the ambiguous instances with prediction set sizes greater than 1 and the atypical instances with the empty prediction set in the set-valued prediction paradigm.

**Multi-armed Bandit (MAB):** The MAB framework (Lai & Robbins, 1985; Auer et al., 2002) concerns a decision-making scenario where a learner sequentially pulls an arm $A$ from a set $\{1, \ldots, K\}$ to maximize cumulative rewards over time. To this end, various policies $\pi$ (could be either a probability distribution or deterministic rule) are proposed to guide the arm pulling, e.g., epsilon-greedy (Sutton & Barto, 2018), Upper Confidence Bound (Auer et al., 2002), and Thompson sampling strategies (Thompson, 1933), etc. The policy $\pi$ and the pulled arm $A$ in the MAB can be conceptually treated as a form of decision rule and a predicted label in the classification problem.

MAB problems are prevalent in various fields, such as online advertising, clinical trials, and recommendation systems, where a context $\boldsymbol{X}$ informs personalized decision-making. Specifically, after receiving the context $\boldsymbol{X}$ from an environment, a learner pulls an arm $A$ that follows $\pi(\cdot \mid \boldsymbol{X})$, and then receives a feedback $\mathbb{1}\{A = Y\}$ returned by the environment. Such bandit feedback only indicates the correctness of the pulled arm. If the feedback is 0, the learner does not know the ground truth and hence faces a challenge in optimizing reward strategies. While existing studies (Kakade et al., 2008; Wang et al., 2010; Crammer & Gentile, 2013; Abbasi-Yadkori et al., 2011; Gollapudi et al., 2021; van der Hoeven et al., 2021) have explored linear and neural network-based (Zhou et al., 2020; Jin et al., 2021; Zhang et al., 2021; Xu et al., 2022) approaches to the bandit problem, they seldom address set-valued classification and/or OOD detection within this framework.

Notably, while research (Taufiq et al., 2022; Zhang et al., 2023; Stanton et al., 2023) has explored set-valued prediction in reinforcement learning, their focus diverges from our bandit-centric investigation. The recent work BCCP (Wang & Qiao, 2024b) is most aligned with our work, yet it does not directly consider the task of OOD detection, a practical requirement in applications. For instance, in social content moderation, a system needs to flag user-generated content into some categories (safe, offensive, etc), and detect new types of harmful content as well. Our study, therefore, introduces a novel approach, considering set-valued classification with OOD detection in the context of bandit feedback, addressing a significant gap in the literature.

**Set-valued Prediction with Bandit Feedback:** The learning framework operates under bandit feedback, where only correctness information for a selected action is observed. At each time step $t$, given a context $\boldsymbol{X}_t$, a learner first produces a set-valued prediction $\phi_t(\boldsymbol{X}_t)$ with a certain theoretical guarantee, indicating a subset of potential labels. Subsequently, a learner takes an action $A_t$ (denoting a single label) according to the policy $\pi_t$, where $A_t$ is sampled from all possibilities. The environment provides a binary feedback signal that whether the pulled arm $A_t$ confirms the ground true label of $\boldsymbol{X}_t$ (see Algorithm 1).

## 3 Methodology

In this section, we develop and optimize our set-valued classifier, BanditGPS, for the online bandit feedback environment. BanditGPS is engineered to accomplish three critical objectives: (1) generate small/informative prediction sets; (2) proficiently detect OOD queries; and (3) maintain the recall for each normal class to be above a threshold of $1 - \gamma$, where $\gamma$ is a tolerance level prescribed by practitioners. The algorithmic framework and operational flow of BanditGPS are outlined in Algorithm 1. Throughout the article, we use $[K]$ to denote the set $\{1, \ldots, K\}$ for convenience.

Consider a sequence of i.i.d. samples with inaccessible labels $(\boldsymbol{X}_t, Y_t) \in \mathcal{X} \times \mathcal{Y}, t = 1, \cdots, T$ generated from the environment. Within this setting, we assume $K$ established normal classes known from historical data. Nonetheless, we anticipate the emergence of OOD instances over time, which may not fit within these predefined classes and potentially signify novel, yet-to-be-identified classes.

To navigate this task, we leverage a neural network as our hypothesis class $\mathcal{F}$. Specifically, we define a decision function vector $\boldsymbol{f}_{\mathcal{W}^t} \in \mathcal{F}$, parameterized by weights $\mathcal{W}^t$ at time $t$, where $\boldsymbol{f}_{\mathcal{W}^t}(\boldsymbol{X}) := (f^1_{\mathcal{W}^t}(\boldsymbol{X}), \cdots, f^K_{\mathcal{W}^t}(\boldsymbol{X}))^\top \in \mathbb{R}^K$ encapsulates the decision functions for each of the $K$ normal classes. For any given query $\boldsymbol{X}_t$ revealed at time $t$, the prediction set is constructed as,

$$\phi_t(\boldsymbol{X}_t) := \left\{ k \in [K] : f^k_{\mathcal{W}^t}(\boldsymbol{X}_t) \geq 0 \right\}. \tag{1}$$

Intuitively speaking, class $k$ is considered to be a plausible label for query $\boldsymbol{X}_t$ if the query falls into the acceptance region for class $k$ defined by $\{\boldsymbol{x} : f^k_{\mathcal{W}^t}(\boldsymbol{x}) \geq 0\}$. Note that the union of all the $K$ many acceptance regions may not cover the entire space $\mathcal{X}$, and regions may overlap with each other.

**Prediction Set Size Minimization:** Based on the above definition of the prediction set, the prediction set size for the query $\boldsymbol{X}_t$ is naturally quantified as $|\phi_t(\boldsymbol{X}_t)| := \sum_{k=1}^K \mathbb{1}\{f^k_{\mathcal{W}^t}(\boldsymbol{X}_t) \geq 0\}$, which ranges from 0 to $K$. In particular, size 0 denotes that the classifier believes $\boldsymbol{X}_t$ is an OOD instance, indicating its distinctiveness from all normal classes. Conversely, size $K$ signifies the maximal ambiguity associated with the query, suggesting that it blurs the lines across all normal classes.

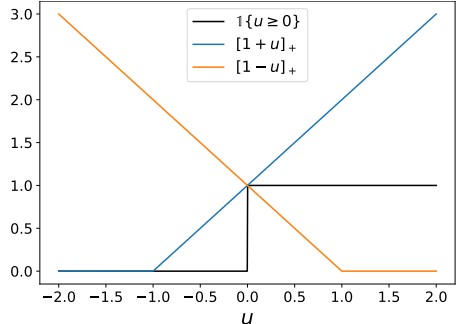

Figure 1: Surrogate loss functions

Due to the discontinuous, and hence, computationally intractable 0-1 loss $\mathbb{1}\{u \geq 0\}$ involved in the definition of prediction set size, we employ a surrogate hinge loss $\ell_1(u) := [1+u]_+ = \max(0, 1+u)$ as a proxy for the 0-1 loss (see Figure 1). This substitution enables a practical way to return a smaller (more informative) prediction set by minimizing the loss

$$\sum_{k=1}^K \ell_1(f^k_{\mathcal{W}^t}(\boldsymbol{X}_t)) = \sum_{k=1}^K [1 + f^k_{\mathcal{W}^t}(\boldsymbol{X}_t)]_+. \tag{2}$$

**OOD Detection Maximization:** A query instance $\boldsymbol{X}_t$ is not flagged as OOD if and only if its prediction set is non-empty, i.e., $|\phi_t(\boldsymbol{X}_t)| > 0$. This is equivalent to the existence of at least one class $k \in [K]$ such that $f^k_{\mathcal{W}^t}(\boldsymbol{X}_t) \geq 0$, or equivalently, $\max_{k \in [K]} f^k_{\mathcal{W}^t}(\boldsymbol{X}_t) \geq 0$. Therefore, to enhance the OOD detection performance, it is imperative to penalize the indicator function of the occurrence of this event, $\mathbb{1}\{\max_{k \in [K]} f^k_{\mathcal{W}^t}(\boldsymbol{X}_t) \geq 0\}$. Given the similar challenge imposed by the 0-1 loss as mentioned above, to effectively tackle the OOD detection task, we instead minimize the following:

$$\ell_1\big(\max_{k \in [K]} f^k_{\mathcal{W}^t}(\boldsymbol{X}_t)\big) = \big[1 + \max_{k \in [K]} f^k_{\mathcal{W}^t}(\boldsymbol{X}_t)\big]_+. \tag{3}$$

**Recall Control for Normal Classes:** By relating the definition of prediction set (1), the class-specific recall is further expressed as $\mathbb{P}\big[Y_t \in \phi_t(\boldsymbol{X}_t) \mid Y_t = k\big] = \mathbb{P}\big[f^k_{\mathcal{W}^t}(\boldsymbol{X}_t) \geq 0 \mid Y_t = k\big]$. Thus, controlling the class-specific normal class recall to the prescribed value at least $1 - \gamma$, that is,

$$\mathbb{P}\big[f^k_{\mathcal{W}^t}(\boldsymbol{X}_t) \geq 0 \mid Y_t = k\big] = \mathbb{E}\big[\mathbb{1}\{f^k_{\mathcal{W}^t}(\boldsymbol{X}_t) \geq 0\} \mid Y_t = k\big] \geq 1 - \gamma, \;\; \text{for } k \in [K]$$

is equivalent to $\mathbb{E}\big[\mathbb{1}\{Y_t = k\} \cdot \big[\mathbb{1}\{f^k_{\mathcal{W}^t}(\boldsymbol{X}_t) < 0\} - \gamma\big]\big] \leq 0$, for $k \in [K]$ due to the Bayes theorem and some algebra (see details in Appendix A.2). This inequality can be viewed as a constraint in constrained optimization problems. Equivalently, our objective function incorporates the left-hand side of this constraint as additional regularization terms, that is,

$$\mathbb{E}\big[\mathbb{1}\{Y_t = k\} \cdot \big[\mathbb{1}\{f^k_{\mathcal{W}^t}(\boldsymbol{X}_t) < 0\} - \gamma\big]\big], \;\; \text{for } k \in [K]. \tag{4}$$

In the bandit feedback setting, the ground truth $Y_t$ and hence the quantities $\mathbb{1}\{Y_t = k\}, k \in [K]$ are unobservable since the learner has no direct access to the ground truth of the label $Y_t$ once receives the query $\boldsymbol{X}_t$. Instead, it only knows whether its pulled arm $A_t$ based on a policy $\pi_t$ is correct or not, i.e., the feedback $\mathbb{1}\{A_t = Y_t\}$. Particularly, if the feedback is 0, the learner has no idea which label is the true class. To overcome this challenge, in practice, the ground truth $\mathbb{1}\{Y_t = k\}$ (the first indicator function in (4)) can be replaced by an unbiased estimation conditional on $(\boldsymbol{X}_t, Y_t)$,

$$\Delta_{t,k} := \frac{\mathbb{1}\{A_t = k\}}{\pi_t(k \mid \boldsymbol{X}_t)} \mathbb{1}\{A_t = Y_t\}, \tag{5}$$

due to the fact $\mathbb{E}_{\pi_t}[\Delta_{t,k} \mid \boldsymbol{X}_t, Y_t] = \mathbb{1}\{Y_t = k\}$, where $\pi_t$ governs the distribution of $A_t$ given $\boldsymbol{X}_t$. Different from the $K$-arm policy, with arms from $[K]$ reviewed in Section 2, we utilize an augmented policy with an additional arm, namely OOD, corresponding to the case that the instance comes from the OOD class. The definition of policy $\pi_t$ is deferred to the next subsection.

To the same token, the intractable 0-1 loss $\mathbb{1}\{u < 0\}$ in the second indicator function in (4) can be substituted by the surrogate hinge loss $[1 - u]_+ := \max(0, 1 - u)$. This amounts to replacing $\mathbb{1}\{u < 0\} - \gamma$ in (4) by $\ell_{2,\gamma}(u) := [1 - u]_+ - \gamma$. Thus, instead of working with the loss function in (4), we can minimize the empirical average of the following loss function in practice:

$$\Delta_{t,k} \cdot \ell_{2,\gamma}(f^k_{\mathcal{W}^t}(\boldsymbol{X}_t)) := \Delta_{t,k} \cdot \big[\big[1 - f^k_{\mathcal{W}^t}(\boldsymbol{X}_t)\big]_+ - \gamma\big]. \tag{6}$$

**Policy Design:** In this article, our policy $\pi_t$ is constructed based on the scoring function $\boldsymbol{f}_{\mathcal{W}^t}(\boldsymbol{x}) = (f^1_{\mathcal{W}^t}(\boldsymbol{x}), \cdots, f^K_{\mathcal{W}^t}(\boldsymbol{x}))^\top$ associated with the normal classes. Specifically, the policy is defined as,

$$\pi_t(k \mid \boldsymbol{X}_t) := \frac{\exp(f^k_{\mathcal{W}^t}(\boldsymbol{X}_t))}{\exp(-\max_{j \in [K]} f^j_{\mathcal{W}^t}(\boldsymbol{X}_t)) + \sum_{j=1}^K \exp(f^j_{\mathcal{W}^t}(\boldsymbol{X}_t))}, \;\; k \in [K],$$

$$\pi_t(\text{OOD} \mid \boldsymbol{X}_t) := \frac{\exp(-\max_{j \in [K]} f^j_{\mathcal{W}^t}(\boldsymbol{X}_t))}{\exp(-\max_{j \in [K]} f^j_{\mathcal{W}^t}(\boldsymbol{X}_t)) + \sum_{j=1}^K \exp(f^j_{\mathcal{W}^t}(\boldsymbol{X}_t))}.$$

Note that $\pi_t(a \mid \boldsymbol{X}_t)$, $a \in \{1, \ldots, K, \text{OOD}\}$, sum up to 1 and hence is a legitimate probability distribution for $A_t$. Under this policy, the probability of pulling arm $k$ is proportional to the exponentiated score for class $k$,

while the probability of pulling OOD is proportional to the exponentiated negative maximum score over all normal classes. Hence, a small score across all normal classes indicates a strong likelihood of the query being from the OOD class.

**The Overall Objective Function:** Finally, taking into consideration three goals, i.e., minimizing prediction set size (2), enhancing OOD detection (3), and controlling class-specific recall (6) for normal classes, the unified risk function for each data instance that we aim to minimize is

$$\mathcal{L}(\boldsymbol{X}_t; \mathcal{W}^t) := \frac{\lambda}{K} \sum_{k=1}^{K} \ell_1(f_{\mathcal{W}^t}^k(\boldsymbol{X}_t)) + (1-\lambda)\ell_1(\max_{k \in [K]} f_{\mathcal{W}^t}^k(\boldsymbol{X}_t)) + \sum_{k=1}^{K} \lambda_{t,k}\Delta_{t,k}\ell_{2,\gamma}(f_{\mathcal{W}^t}^k(\boldsymbol{X}_t)), \qquad (7)$$

where the neural network $\boldsymbol{f}_{\mathcal{W}^t}$ is updated by the stochastic gradient descent (or its variants). Here $\lambda \in [0,1]$ is a user-specified value to indicate the trade-off between prediction set size and OOD detection performance. Note that we do not treat $\lambda$ as a tuning parameter and rather leave it to be specified by the user according to their unique needs.

As for the parameter $\lambda_{t,k}$, an improperly chosen $\lambda_{t,k}$ may lead to two cases: a $\lambda_{t,k}$ that is too small might fail to control the recall for normal classes, while a $\lambda_{t,k}$ that is too large can cause the empirical normal recall highly above the pre-requisite $1-\gamma$ and hence enlarge the prediction sets and impair the OOD detection performance. Additionally, an appropriate value of $\lambda_{t,k}$ can vary across tasks, making it difficult to predefine an optimal value. To address these limitations, we employ the primal-dual algorithm to dynamically optimize the value of $\lambda_{t,k}$ (see Equation (8) in Algorithm 1). Specifically, if the current prediction is over control, i.e., $\ell_{2,\gamma}(f_{\mathcal{W}^t}^k(\boldsymbol{X}_t)) < 0$, Equation (8) will adaptively lead to a smaller value of the parameter $\lambda_{t+1,k}$ in the next iteration. Such

---

**Algorithm 1** BanditGPS

**Require:** Initialize weight matrices $\mathcal{W}^1$ and $\lambda_{1,k}$, $k \in [K]$. Given $\lambda$ and learning rates $\eta_1, \eta_2$.
1: **for** $t = 1, 2, 3, \ldots, T$ **do**
2:      Learner receives $\boldsymbol{X}_t$ and returns a prediction set

$$\phi_t(\boldsymbol{X}_t) := \left\{ k \in [K] : f_{\mathcal{W}^t}^k(\boldsymbol{X}_t) \geq 0 \right\}$$

3:      Learner pulls an arm $A_t \sim \pi_t$ and computes $\Delta_{t,k}$
4:      Update parameters:

$$\begin{cases} \mathcal{W}^{t+1} = \mathcal{W}^t - \eta_1 \nabla_{\mathcal{W}} \mathcal{L}(\boldsymbol{X}_t; \mathcal{W}^t) \\ \lambda_{t+1,k} = \left[ \lambda_{t,k} + \eta_2 \Delta_{t,k} \ell_{2,\gamma}(f_{\mathcal{W}^t}^k(\boldsymbol{X}_t)) \right]_+ \end{cases} \quad (8)$$

5: **end for**

---

adjustment helps potentially return smaller prediction sets. Similarly, when $\ell_{2,\gamma}(f_{\mathcal{W}^t}^k(\boldsymbol{X}_t)) > 0$, Equation (8) will lead to a larger value of $\lambda_{t+1,k}$ in the next iteration. This automatic updating rule frees us from the burden of manually selecting the value of $\lambda_{t+1,k}$ with excessive effort.

## 4 Main Theorems

This section provides the convergence rate analysis concerning the regret associated with prediction set size and OOD detection performance. Our analysis builds upon foundational work in over-parameterized neural networks (Du et al., 2019; Allen-Zhu et al., 2019; Cao & Gu, 2019; 2020; Chen et al., 2023), i.e., sufficiently wide neural networks.

Let the hypothesis class of ReLU neural networks with depth $L$ and a constant width $m$ be

$$\mathcal{F} := \left\{ \boldsymbol{f}_{\mathcal{W}} : \mathbb{R}^p \mapsto \mathbb{R}^K \mid \boldsymbol{f}_{\mathcal{W}}(\cdot) = \mathbf{W}_L \sigma_{L-1}\big(\mathbf{W}_{L-1}\sigma_{L-2}\big(\cdots \sigma_1(\mathbf{W}_1(\cdot))\big) \right\},$$

where $\mathcal{W} := (\text{Vec}(\mathbf{W}_1)^\top, \cdots, \text{Vec}(\mathbf{W}_L)^\top)^\top$ denotes a concatenated long vector by vectorizing $\mathbf{W}_1 \in \mathbb{R}^{m \times p}, \mathbf{W}_l \in \mathbb{R}^{m \times m}$ for $2 \leq l \leq L-1$, and $\mathbf{W}_L \in \mathbb{R}^{K \times m}$. The ReLU activation functions $\sigma_l, l \in [L-1]$ are element-wisely applied on each layer. Additionally, we define the ball around the neural network's initialization as $\mathcal{B}(\mathcal{W}^1, \omega) := \{\mathcal{W} : \|\mathbf{W}_l - \mathbf{W}_l^1\|_2 \leq \omega, l \in [L]\}$. In this article, the Frobenius norm of a matrix is denoted by $\|\cdot\|_2$ and this notation extends to tensors.

Consider a restricted hypothesis class that adheres to the empirical normal recall control within this ball whose radius scales in the order of $m^{-1/2}$:

$$\mathcal{F}^+ := \left\{ \boldsymbol{f}_{\mathcal{W}} \in \mathcal{F} : \frac{1}{T} \sum_{t=1}^{T} \mathbb{1}\{Y_t = k\} \cdot \ell_{2,\gamma}(f_{\mathcal{W}}^k(\boldsymbol{X}_t)) \leq 0, k \in [K] \text{ and } \mathcal{W} \in \mathcal{B}(\mathcal{W}^1, Rm^{-1/2}) \right\},$$

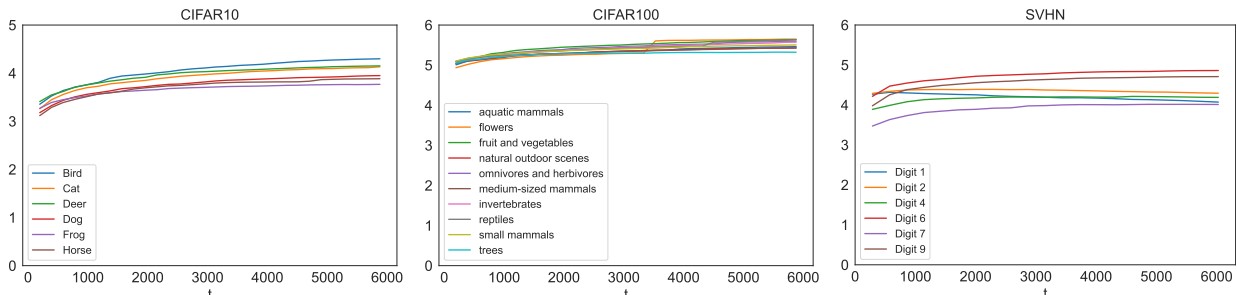

Figure 2: Updated $\lambda_{t,k}$ for normal class across three datasets when $\lambda = 1$

where $R > 0$ is a fixed constant. We define the loss function as $\ell(\boldsymbol{f}_{\mathcal{W}}(\boldsymbol{X})) := \frac{\lambda}{K} \sum_{k=1}^{K} \ell_1(f_{\mathcal{W}}^k(\boldsymbol{X})) + (1 - \lambda)\ell_1(\max_{k \in [K]} f_{\mathcal{W}}^k(\boldsymbol{X}))$ which consists of the loss on prediction set size and the loss on OOD detection; they may be viewed as a negative reward and both are parts of the objective function $\mathcal{L}(\boldsymbol{X}; \mathcal{W})$ in Equation (7). In this context, regret is defined as the difference between the loss achieved by the algorithm and the least loss that could have been achieved by the best possible action taken in every iteration. To derive the rate of convergence for the regret, we use the following three assumptions.

**Assumption 1.** We assume the entries in $\mathbf{W}_l^1, l \in [L-1]$ are initialized with Gaussian distribution $\mathcal{N}(0, \frac{2}{m})$ and the entries in $\mathbf{W}_L^1$ are initialized with Gaussian distribution $\mathcal{N}(0, \frac{1}{K})$.

**Assumption 2.** The instance $\boldsymbol{X}_t \in \mathbb{R}^p, t \in [T]$ is normalized to have a unit norm, i.e., $\|\boldsymbol{X}_t\|_2 = 1$.

**Assumption 3.** $\exists\, C_\lambda, C_\Delta > 0$, s.t. for each $k \in [K], t \in [T]$, $\lambda_{t,k} \leq C_\lambda, \Delta_{t,k} \leq C_\Delta$.

*Remark* 4. Assumption 1 and Assumption 2 are standard assumptions in the regime of over-parameterized neural networks. Cao & Gu (2020) extends Assumption 2 to the one with bound norms.

The assumption of the bounded $\lambda_{t,k}$ is empirically verified as in Figure 2 for the case of $\lambda = 1$. It shows that $\lambda_{t,k} \leq 6$ holds for all $t$ and $k$ across the three datasets we work with. The assumption of the boundness of $\Delta_{t,k}$ is not strong either because its upper bound is inversely related to the lower bound of $\pi_t(k \mid \boldsymbol{X}_t) \neq 0$ (see Equation (5)), which can be manually manipulated in practice.

**Theorem 5.** *Let $p_k := \mathbb{P}[Y_t = k]$ be the prior probability for normal class $k \in [K]$. The average recall for any class $k$ over all time for the prediction sets returned by Algorithm 1 is guaranteed:*

$$\frac{1}{T} \sum_{t=1}^{T} \mathbb{P}\left[Y_t \in \phi_t(\boldsymbol{X}_t) \mid Y_t = k\right] \geq 1 - \gamma - \frac{C_\lambda}{T \eta_2 p_k} \; \forall k.$$

The above probability is taken over all the randomness during the learning process. If the learning rate $\eta_2 = \mathcal{O}(T^{-\alpha})$ with $\alpha \in [0, 1)$, Theorem 5 implies that, on average, the accumulative normal recall approaches to the prescribed value $1 - \gamma$ with rate $\mathcal{O}(T^{\alpha-1})$. Note that the recall bound may be less meaningful in a short-term regime. Intuitively, if $p_k$ is very small, it means that we may not collect sufficient samples for class $k$ within a finite time horizon, leading to potential failures in recall control. However, the denominator in Theorem 5 includes $T$, compensating for this issue in the long run.

**Theorem 6.** *Define an optimal learner with controlled normal recalls from the hypothesis class $\mathcal{F}^+$ as $\boldsymbol{f}_{\mathcal{W}^*} := \arg\min_{\boldsymbol{f}_{\mathcal{W}} \in \mathcal{F}^+} \frac{1}{T} \sum_{t=1}^{T} \ell(\boldsymbol{f}_{\mathcal{W}}(\boldsymbol{X}_t))$. There exists an absolute constant $\kappa$ satisfying that, for any constant $R > 0$, $m \geq \mathcal{O}(\kappa^{-2} R^2 L^{12}[\log(m)]^3)$, and the learning rate $\eta_1 = \mathcal{O}(\frac{R}{\sqrt{KTm}})$, with probability at least $1 - \mathcal{O}(TL^2)\exp(-\Omega(m\omega^{2/3}L))$ over the randomness of the initialization $\mathcal{W}^1$, the regret is bounded as*

$$Regret(T) := \frac{1}{T} \sum_{t=1}^{T} \ell(\boldsymbol{f}_{\mathcal{W}^t}(\boldsymbol{X}_t)) - \frac{1}{T} \sum_{t=1}^{T} \ell(\boldsymbol{f}_{\mathcal{W}^*}(\boldsymbol{X}_t)) \leq c^* \mathcal{O}\left(\frac{LR\sqrt{K}}{\sqrt{T}} + \frac{L^3 R^{4/3} \sqrt{\log(m)}}{m^{1/6}}\right),$$

*where the constant $c^* := (\frac{\lambda}{\sqrt{K}} + 1 - \lambda + \sqrt{K} C_\lambda C_\Delta)^2 + \frac{\lambda}{K} + \frac{1-\lambda}{\sqrt{K}} + K C_\lambda C_\Delta$.*

Theorem 6 shows that when the width of the neural network is sufficiently large, Algorithm 1 leads to the regret convergence rate with the order $\mathcal{O}(T^{-1/2} + m^{-1/6}\sqrt{\log(m)})$. Particularly, if the width further satisfies

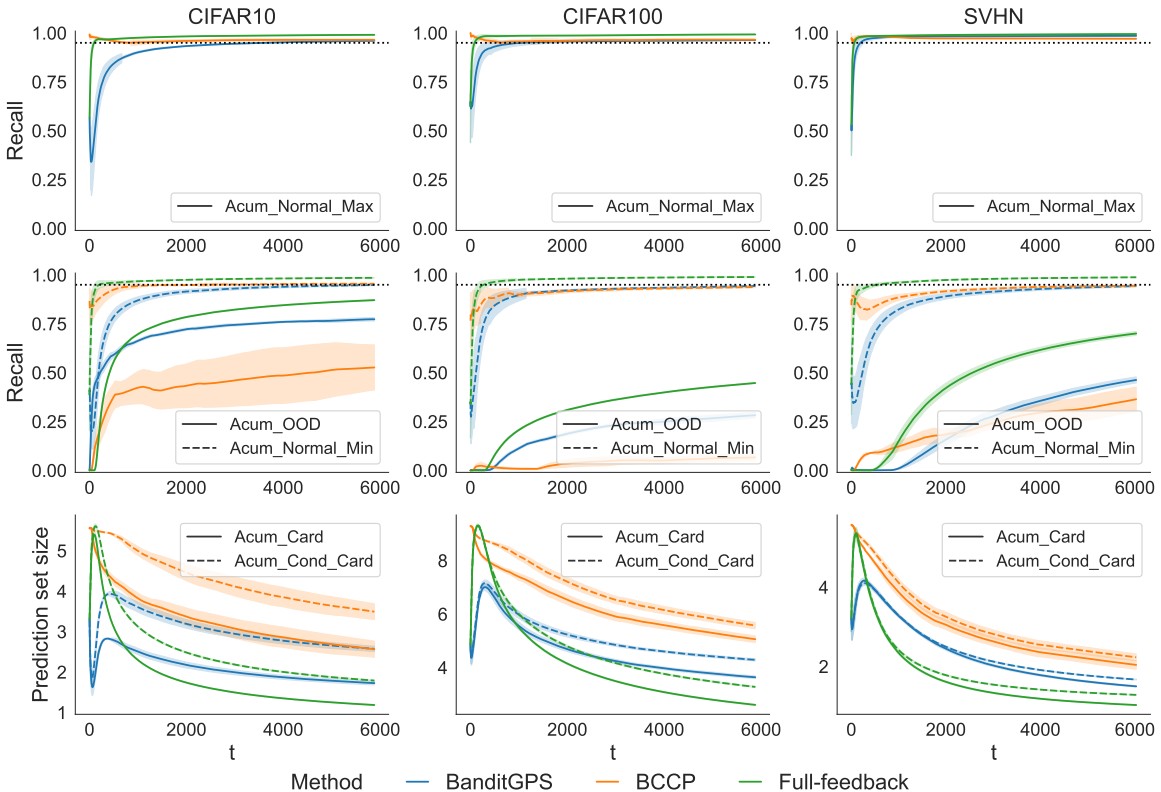

Figure 3: Comparison among methods based on accumulative performance.

$m \geq \mathcal{O}(T^3)$, then the regret is close to the parametric rate $\tilde{\mathcal{O}}(T^{-1/2})$, which is similar to the convex online learning. Similar convergence rate results were previously shown in Allen-Zhu et al. (2019); Cao & Gu (2019); Chen et al. (2023) despite different settings.

## 5 Experiments

**Baselines:** Due to the unique setting and task, BCCP (Wang & Qiao, 2024b) represents the most aligned approach to our context even though it lacks mechanisms for OOD detection. To establish a baseline for comparison, we extend BCCP to also output an empty set, indicating OOD instances. Additionally, we present the performance of our model under the full feedback scenario, where the model is trained based on the labeled data, in order to show the performance in an ideal setting as a benchmark.

**Set-up:** We compare the methods by evaluating them on CIFAR10, CIFAR100, and SVHN datasets. For CIFAR10, we set {Bird, Cat, Deer, Dog, Frog, Horse} as normal classes while all the remaining {Airplane, Car, Ship, Truck} as the OOD class; for CIFAR100, we treat 10 coarser classes {Aquatic mammals, Flowers, Fruit and vegetables, Natural outdoor scenes, Omnivores and herbivores, Medium-sized mammals, Invertebrates, Reptiles, Small mammals, Trees} as normal classes, and the remaining 10 coarser classes as the entire OOD class; for SVHN, we let Digits {1, 2, 4, 6, 7, 9} as the normal classes and the remaining Digits {0, 3, 5, 8} be the OOD superclass.

We let all the experiments have the same desired recall $1 - \gamma = 0.95$ across datasets, utilizing ResNet (He et al., 2016) as the backbone architecture, Adam for optimization, learning rate $\eta_1 = 10^{-4}$ for network updates, and $\eta_2 = \eta_2(t) = t^{-1/2}$ for optimizing $\lambda_{t,k}$. To improve the computational efficiency, model updates employ batch data with a size of 256 in each iteration, with about 6000 total iterations.

**Metrics:** For each iteration $t \in [T]$, we report several accumulative quantities (see definitions in Table 1): Acum_OOD$(t)$, Acum_Normal_Min$(t)$, Acum_Normal_Max$(t)$ show the OOD detection performance and the recall control on normal classes while Acum_Card$(t)$, Acum_Cond_Card$(t)$ assess the prediction set

Table 1: Displayed Metrics for Comparison

| Accumulative metric | Metric on the hold-out dataset |
|---|---|
| $\text{Acum\_Normal\_Min}(t) = \min_{k \in [K]} \text{Acum\_recall}(t, k)$ | $\text{Hdt\_Normal\_Min}(t) = \min_{k \in [K]} \text{Hdt\_recall}(t, k)$ |
| $\text{Acum\_Normal\_Max}(t) = \max_{k \in [K]} \text{Acum\_recall}(t, k)$ | $\text{Hdt\_Normal\_Max}(t) = \max_{k \in [K]} \text{Hdt\_recall}(t, k)$ |
| $\text{Acum\_OOD}(t) = \text{Acum\_recall}(t, \text{OOD})$ | $\text{Hdt\_OOD}(t) = \text{Hdt\_recall}(t, \text{OOD})$ |
| $\text{Acum\_Cond\_Card}(t) = \dfrac{\sum_{s=1}^{t} \sum_{\mathbf{X}_i \in \mathcal{B}_s} |\phi_t(\mathbf{X}_i)| \cdot \mathbb{1}\{Y_i \neq \text{OOD}\}}{\sum_{s=1}^{t} \sum_{\mathbf{X}_i \in \mathcal{B}_s} \mathbb{1}\{Y_i \neq \text{OOD}\}}$ | $\text{Hdt\_Cond\_Card}(t) = \dfrac{\sum_{i=1}^{n} |\phi_t(\mathbf{X}_i)| \cdot \mathbb{1}\{Y_i \neq \text{OOD}\}}{\sum_{i=1}^{n} \mathbb{1}\{Y_i \neq \text{OOD}\}}$ |
| $\text{Acum\_Card}(t) = \dfrac{\sum_{s=1}^{t} \sum_{\mathbf{X}_i \in \mathcal{B}_s} |\phi_t(\mathbf{X}_i)|}{\sum_{s=1}^{t} |\mathcal{B}_s|}$ | $\text{Hdt\_Card}(t) = \frac{1}{n} \sum_{i=1}^{n} |\phi_t(\mathbf{X}_i)|$ |

size. Additionally, we assess classifiers' performance on a fixed holdout labeled dataset after each iteration. In Table 1, $\mathcal{B}_s$ denotes the batch of data at iteration $s$, and $n$ denotes the sample size of the holdout dataset, and

$$\text{Acum\_recall}(t, k)$$
$$:= \frac{\sum_{s=1}^{t} \sum_{\mathbf{X}_i \in \mathcal{B}_s} \mathbb{1}\{Y_i = k \ \& \ Y_i \in \phi_t(\mathbf{X}_i)\}}{\sum_{s=1}^{t} \sum_{\mathbf{X}_i \in \mathcal{B}_s} \mathbb{1}\{Y_i = k\}}, \quad k \in [K]$$

$$\text{Acum\_recall}(t, \text{OOD})$$
$$:= \frac{\sum_{s=1}^{t} \sum_{\mathbf{X}_i \in \mathcal{B}_s} \mathbb{1}\{Y_i = \text{OOD} \ \& \ |\phi_t(\mathbf{X}_i)| = 0\}}{\sum_{s=1}^{t} \sum_{\mathbf{X}_i \in \mathcal{B}_s} \mathbb{1}\{Y_i = \text{OOD}\}}$$

$$\text{Hdt\_recall}(t, k)$$
$$:= \frac{\sum_{i=1}^{n} \mathbb{1}\{Y_i = k \ \& \ Y_i \in \phi_t(\mathbf{X}_i)\}}{\sum_{i=1}^{n} \mathbb{1}\{Y_i = k\}}, \quad k \in [K]$$

$$\text{Hdt\_recall}(t, \text{OOD})$$
$$:= \frac{\sum_{i=1}^{n} \mathbb{1}\{Y_i = \text{OOD} \ \& \ |\phi_t(\mathbf{X}_i)| = 0\}}{\sum_{i=1}^{n} \mathbb{1}\{Y_i = \text{OOD}\}}$$

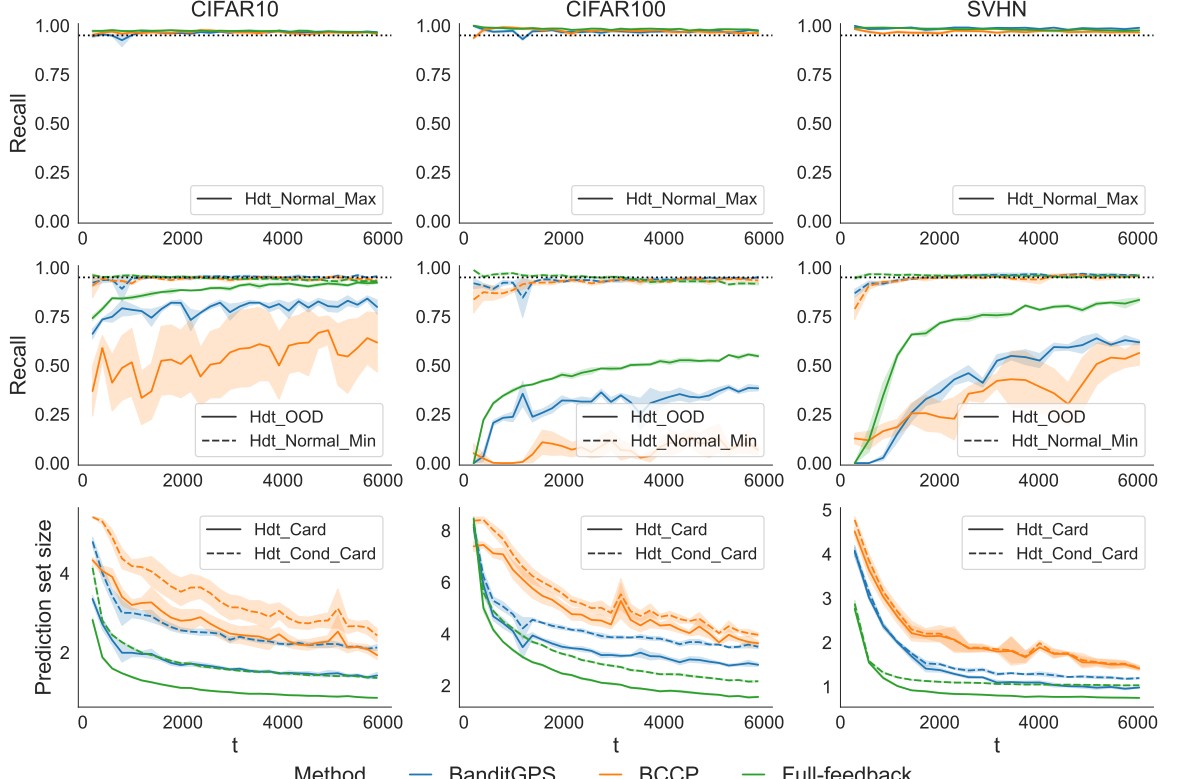

Figure 4: Comparison among methods based on performance on the fixed holdout datasets.

Particularly, the first two rows in Table 1 show the extreme cases of recall control on normal classes in each iteration. The third row displays the OOD detection performance. In terms of prediction informativeness, the fourth row reports the prediction set size exclusively on all normal classes while the last row presents the prediction set size on all observations, including those from the OOD class.

**Results:** For a fair comparison with the BCCP where OOD detection is not particularly targeting, we first set $\lambda = 1$ in the risk function $\mathcal{L}(\cdot, \cdot)$ of BanditGPS, implying that the optimization of the OOD section performance is not a priority either. The top panel in Figure 3 demonstrates that both BanditGPS and BCCP effectively manage the accumulative class-specific normal recall, as illustrated by the colored dashed curves showing the smallest recall over all normal classes, which approach $1 - \gamma$ as iterations progress. On the other hand, BanditGPS distinguishes itself from BCCP with superior OOD detection capabilities (solid curves in the top panel) and smaller prediction sets (bottom panel). Further assessment on a holdout labeled dataset (see Figure 4) reinforces BanditGPS's enhanced performance over BCCP, showcasing its robust and improved OOD detection and informative prediction set. With the full feedback information, metrics in Figures 3 and 4 quickly improve as shown in green curves.

**Trade-offs between Prediction Set Size and OOD Recall Driven by $\lambda$:** The parameter $\lambda$ affects the trade-off between metrics on prediction set size and OOD recall. In this section, we study these two metrics by varying the values of $\lambda$, i.e., $0, 0.25, 0.5, 0.75$, and $1$. As shown in Figures 5 and 6, the smaller the $\lambda$ is, the larger the prediction set will be, despite the improvement in the OOD recall.

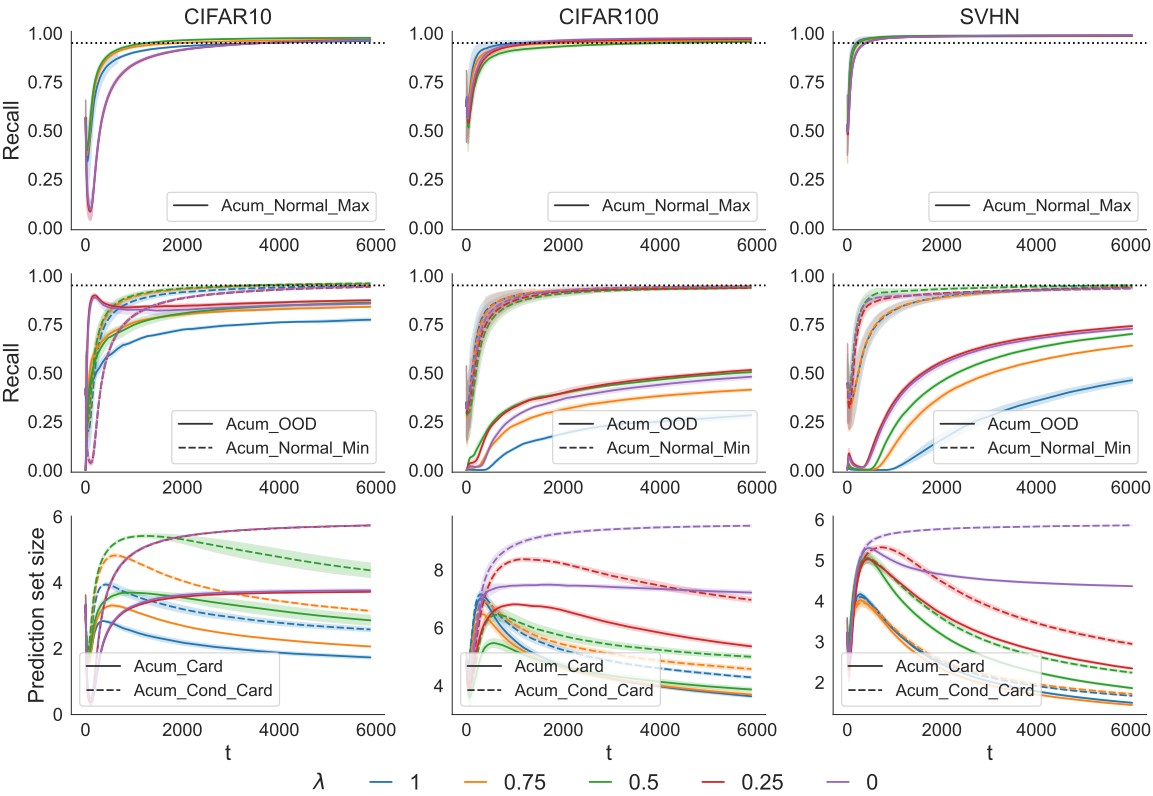

Figure 5: Trade-off among accumulative metrics when varying the value of $\lambda$.

## 6 Discussions

This work introduces a novel online set-valued prediction framework with out-of-distribution (OOD) detection under bandit feedback. While we provide theoretical guarantees and empirical validation, we acknowledge several modeling assumptions and algorithmic components that deserve further clarification and suggest directions for future work.

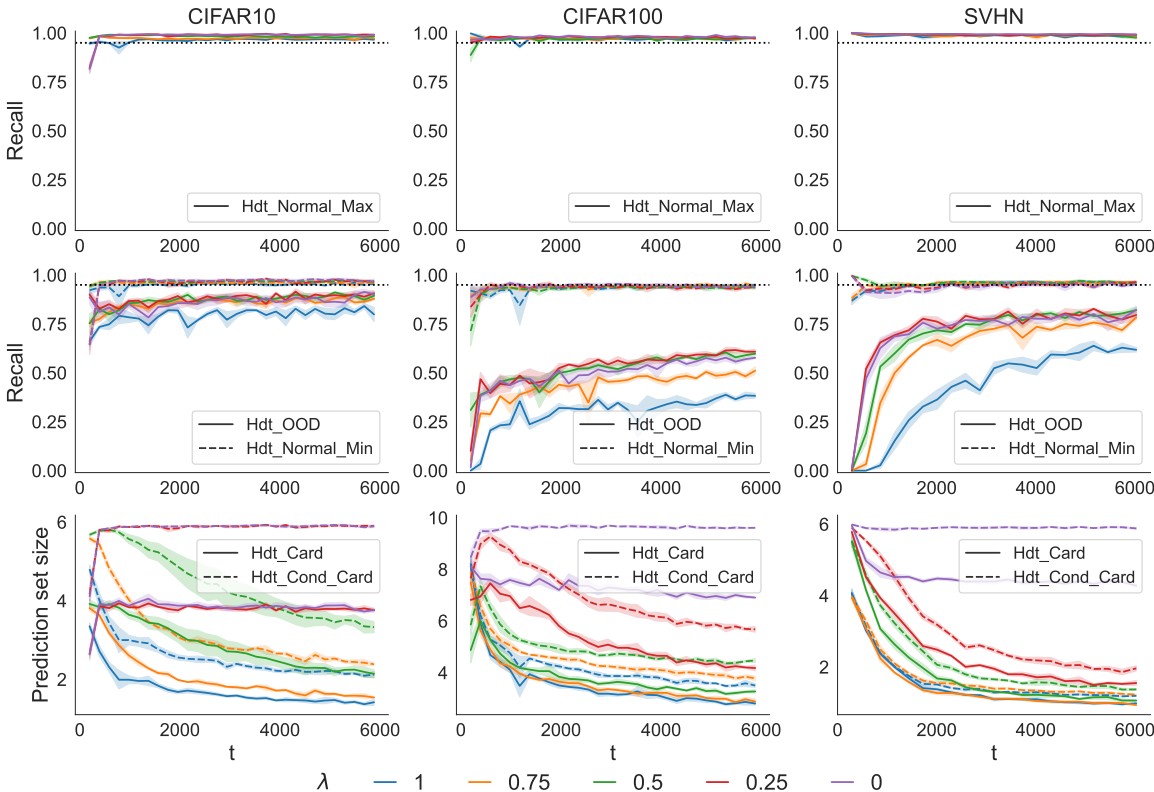

Figure 6: Trade-off among metrics for the fixed holdout dataset when varying the value of $\lambda$.

First, the theoretical analysis in Theorem 6 relies on Assumption 3, which requires the boundedness of both the importance-weighted estimator $\Delta_{t,k}$ and the Lagrange multiplier $\lambda_{t,k}$. In practice, $\Delta_{t,k}$ depends inversely on the action sampling policy $\pi_t(k \mid \boldsymbol{X}_t)$. To ensure $\Delta_{t,k}$ remains bounded, a simple yet effective solution is to clip the policy from below by a constant $\underline{p}$, which guarantees $\Delta_{t,k} \leq \max\{1/\underline{p}, 1/(1 - K\underline{p})\}$. This design consideration can be naturally incorporated into the algorithm to ensure numerical stability and theoretical validity.

On the other hand, the multiplier $\lambda_{t,k}$ is dynamically updated to enforce recall constraints for normal classes. While our method adopts a primal-dual approach, its precise theoretical bound is not given in this work. A promising direction is to build upon frameworks from online constrained optimization (Yu et al., 2017; Yu & Neely, 2020), which offer high-probability or expectation-based bounds for Lagrange multipliers under mild conditions. We plan to incorporate these insights into future extensions of BanditGPS.

Lastly, our current formulation operates in a binary bandit feedback setting, where a single arm is pulled after set-valued prediction. This separates the prediction step (generating a set) from the feedback step (observing a label). An interesting future direction is to unify these steps within a combinatorial bandit framework, where the predicted set itself is treated as the action, and partial or structured feedback is received over the set. Such a generalization could enhance learning efficiency and better reflect real-world applications where multiple predictions are jointly evaluated.

## 7 Conclusions

In the high-risk scenarios exacerbated by uncertainties from ambiguous instances and the emergence of new classes in dynamic environments, we introduced BanditGPS, a novel set-valued classification method tailored for online bandit feedback settings. BanditGPS navigates the uncertainties by offering a set of plausible labels for ambiguous instances and an empty label set for an OOD observation.

BanditGPS handles the challenge of inaccessible label information inherent in bandit feedback by utilizing an estimation of the ground truth of class labels. Leveraging a primal-dual algorithm, BanditGPS dynamically adjusts tuning parameters in response to its performance history, facilitating an adaptive and automatic risk management mechanism. Furthermore, by enabling the weight between informative prediction and OOD detection, BanditGPS empowers practitioners to tailor the system based on their specific operational requirements. The theoretical and empirical validations of BanditGPS affirm its efficacy in managing the challenges of providing informative predictions while accurately detecting OOD instances under the control of class-specific normal recall.

There are limitations in this initial work that warrant further investigation and improvement. Beyond those discussed in Section 6, one promising direction is to integrate BanditGPS with alternative multi-armed bandit strategies that offer theoretical performance guarantees, such as Thompson Sampling or the Upper Confidence Bound algorithm, to further enhance its decision-making capabilities. Additionally, rather than treating all abnormal inputs as belonging to a single super OOD class, adapting BanditGPS to support class-incremental learning could enable more refined discovery and assimilation of newly emerging classes in dynamic environments.

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

# A    Appendix

## A.1    Experiments Compute Resources

All experiments are conducted on an NVIDIA P100 GPU with CUDA 11.3. The empirical metrics presented in all figures are computed and averaged over 4 runs, with the shaded regions representing the standard error.

## A.2    Derive for Normal Recall Control

$$\mathbb{E}\left[\mathbb{1}\{f_{\mathcal{W}^t}^k(\boldsymbol{X}_t) \geq 0\} \mid Y_t = k\right] \geq 1 - \gamma$$
$$\mathbb{E}\left[1 - \mathbb{1}\{f_{\mathcal{W}^t}^k(\boldsymbol{X}_t) \geq 0\} - \gamma \mid Y_t = k\right] \leq 0$$
$$\mathbb{E}\left[\mathbb{1}\{f_{\mathcal{W}^t}^k(\boldsymbol{X}_t) < 0\} - \gamma \mid Y_t = k\right] \leq 0$$
$$\frac{\mathbb{E}\left[\mathbb{1}\{Y_t = k\} \cdot \left[\mathbb{1}\{f_{\mathcal{W}^t}^k(\boldsymbol{X}_t) < 0\} - \gamma\right]\right]}{\mathbb{P}[Y_t = k]} \leq 0$$
$$\mathbb{E}\left[\mathbb{1}\{Y_t = k\} \cdot \left[\mathbb{1}\{f_{\mathcal{W}^t}^k(\boldsymbol{X}_t) < 0\} - \gamma\right]\right] \leq 0$$

## A.3    Proofs

*Proof of Theorem 5:* From the updating rule Equation (8), we have

$$\lambda_{t+1,k} = \left[\lambda_{t,k} + \eta_2 \Delta_{t,k} \ell_{2,\gamma}(f_{\mathcal{W}^t}^k(\boldsymbol{X}_t))\right]_+$$
$$\geq \lambda_{t,k} + \eta_2 \Delta_{t,k} \ell_{2,\gamma}(f_{\mathcal{W}^t}^k(\boldsymbol{X}_t))$$
$$\implies \sum_{t=1}^{T} \Delta_{t,k} \ell_{2,\gamma}(f_{\mathcal{W}^t}^k(\boldsymbol{X}_t)) \leq \frac{1}{\eta_2} \sum_{t=1}^{T} \lambda_{t+1,k} - \lambda_{t,k} \leq \frac{\lambda_{T+1,k}}{\eta_2} \leq \frac{C_\lambda}{\eta_2},$$

By taking the expectation for both sides of the last inequality, we have

$$\sum_{t=1}^{T} \mathbb{E}[\Delta_{t,k} \ell_{2,\gamma}(f_{\mathcal{W}^t}^k(\boldsymbol{X}_t))] \leq \frac{C_\lambda}{\eta_2}$$

$$\implies \sum_{t=1}^{T} \mathbb{E}\left[\mathbb{E}[\Delta_{t,k} \ell_{2,\gamma}(f_{\mathcal{W}^t}^k(\boldsymbol{X}_t)) \mid \{\pi_s\}_{s<t}, \{(\boldsymbol{X}_t, Y_t)\}_{s \leq t}, \mathcal{W}^1]\right] \leq \frac{C_\lambda}{\eta_2}$$

$$\implies \sum_{t=1}^{T} \mathbb{E}\left[\mathbb{1}\{Y_t = k\} \cdot \ell_{2,\gamma}(f_{\mathcal{W}^t}^k(\boldsymbol{X}_t))\right] \leq \frac{C_\lambda}{\eta_2} \tag{9}$$

$$\implies \sum_{t=1}^{T} \mathbb{E}\left[\ell_{2,\gamma}(f_{\mathcal{W}^t}^k(\boldsymbol{X}_t)) \mid Y_t = k\right] \leq \frac{C_\lambda}{\eta_2 p_k} \tag{10}$$

$$\implies \sum_{t=1}^{T} \mathbb{E}\left[\mathbb{1}\{f_{\mathcal{W}^t}^k(\boldsymbol{X}_t) < 0\} - \gamma \mid Y_t = k\right] \leq \frac{C_\lambda}{\eta_2 p_k} \tag{11}$$

$$\implies \sum_{t=1}^{T} \mathbb{P}\left[f_{\mathcal{W}^t}^k(\boldsymbol{X}_t) < 0] \mid Y_t = k\right] \leq T\gamma + \frac{C_\lambda}{\eta_2 p_k},$$

where (9) holds due to the (conditional) unbiased estimator $\Delta_{t,k}$; (10) holds due to the Bayes theorem, and (11) holds due to the substitution with the surrogate hinge loss as mentioned in (6). Therefore, the last inequality concludes

$$\frac{1}{T} \sum_{t=1}^{T} \mathbb{P}\left[Y_t \in \phi_t(\boldsymbol{X}_t) \mid Y_t = k\right] \geq 1 - \gamma - \frac{C_\lambda}{T\eta_2 p_k}.$$

□

Our derivation for Theorem 6 highly relies on some commonly used lemmas and propositions in the over-parameterized neural network literature. We skip their proofs to avoid redundancy. We leave their details to the main literature Allen-Zhu et al. (2019); Cao & Gu (2019; 2020) for the reader's interest.

Additionally, consider two sequences $\{a_n\}$ and $\{b_n\}$. We employ some asymptotic notations as follows: $a_n = \mathcal{O}(b_n)$ if $\limsup_{n\to\infty} |\frac{a_n}{b_n}| < \infty$; $a_n = \tilde{\mathcal{O}}(b_n)$ to indicate an additional logarithmic factor within $\mathcal{O}(\cdot)$; and $a_n = \Omega(b_n)$ if $\liminf_{n\to\infty} |\frac{a_n}{b_n}| > 0$.

For a fixed instance $\boldsymbol{X}_t$, the neural network can be re-written in the form of matrix products as below:

$$\boldsymbol{f}_{\mathcal{W}}(\boldsymbol{X}_t) = \mathbf{W}_L \mathbf{D}_{t,L-1} \mathbf{W}_{L-1} \mathbf{D}_{t,L-2} \cdots \mathbf{D}_{t,l} \mathbf{W}_l \mathbf{D}_{t,l-1} \cdots \mathbf{D}_{t,1} \mathbf{W}_1 \boldsymbol{X}_t. \tag{12}$$

We define $\boldsymbol{h}_{t,l} := \mathbf{D}_{t,l} \mathbf{W}_{l-1} \mathbf{D}_{L-2} \cdots \mathbf{D}_{t,1} \mathbf{W}_1 \boldsymbol{X}_t, l \in [L-1]$, where

$$\mathbf{D}_{t,l} = \mathrm{Diag}\left( \mathbb{1}\{(\mathbf{W}_l \boldsymbol{h}_{t,l-1})_1 \geq 0\}, \cdots, \mathbb{1}\{(\mathbf{W}_l \boldsymbol{h}_{t,l-1})_m \geq 0\} \right).$$

Thus, the gradient can be written as

$$\nabla_{\mathbf{W}_l} f_{\mathcal{W}}^k(\boldsymbol{X}_t) = \begin{cases} \mathbf{e}_k \boldsymbol{h}_{t,L-1}^\top & \text{if } l = L \\ \left( \boldsymbol{h}_{t,l-1} \mathbf{e}_k^\top \mathbf{W}_L \left( \prod_{r=l+1}^{L-1} \mathbf{D}_{t,r} \mathbf{W}_r \right) \mathbf{D}_{t,l} \right)^\top & \text{if } l \in [L-1] \end{cases} \tag{13}$$

where $\mathbf{e}_k \in \mathbb{R}^K$ is the column vector with value 1 in $k$-th entry and 0 otherwise. When $\mathcal{W}$ in Equation (12) is particularly set as the initialization $\mathcal{W}^1$, the associated $\mathbf{W}_l, \mathbf{D}_{t,l}$ in Equation (12) will be denoted by $\mathbf{W}_l^1, \mathbf{D}_{t,l}^1$ accordingly.

**Lemma 7.** *If $K \leq \mathcal{O}(\frac{m}{L\log(m)})$, with the probability at least $1 - \exp(-\Omega(m\omega^{2/3}L\log(m)))$ over the randomness of $\mathbf{W}_l^1, l \in [L]$, for any perturbation matrices $\mathbf{W}_l'', l \in [L]$ with $\|\mathbf{W}_l''\|_2 \leq \omega = \mathcal{O}(L^{-1.5})$, any diagonal matrices $\mathbf{D}_{t,l}'' \in [-3,3]^{m\times m}, l \in [L-1]$ with at most $\mathcal{O}(m\omega^{2/3}L)$ non-zero entries, we have*

$$\left\| \prod_{l=a+1}^{L} \left( \mathbf{W}_l^1 + \mathbf{W}_l'' \right) \left( \mathbf{D}_{t,l-1}^1 + \mathbf{D}_{t,l-1}'' \right) - \prod_{l=a+1}^{L} \mathbf{W}_l^1 \mathbf{D}_{t,l-1}^1 \right\|_2 \leq \mathcal{O}\left( \frac{\omega^{1/3} L^2 \sqrt{m\log(m)}}{\sqrt{K}} \right).$$

*Proof of Lemma 7:*

$$\left\| \prod_{l=a+1}^{L} \left( \mathbf{W}_l^1 + \mathbf{W}_l'' \right) \left( \mathbf{D}_{t,l-1}^1 + \mathbf{D}_{t,l-1}'' \right) - \prod_{l=a+1}^{L} \mathbf{W}_l^1 \mathbf{D}_{t,l-1}^1 \right\|_2$$

$$= \left\| \sum_{l=a}^{L-1} \mathbf{W}_L^1 \mathbf{D}_{t,L-1}^1 \cdots \mathbf{W}_{l+1}^1 \mathbf{D}_{t,l}'' \left( \mathbf{W}_l^1 + \mathbf{W}_l'' \right) \cdots \left( \mathbf{D}_{t,a}^1 + \mathbf{D}_{t,a}'' \right) \right.$$

$$\left. - \sum_{l=a+1}^{L} \mathbf{W}_L^1 \mathbf{D}_{t,L-1}^1 \cdots \mathbf{W}_{l+1}^1 \mathbf{D}_{t,l}^1 \mathbf{W}_l'' \left( \mathbf{D}_{t,l-1}^1 + \mathbf{D}_{t,l-1}'' \right) \left( \mathbf{W}_{l-1}^1 + \mathbf{W}_{l-1}'' \right) \cdots \left( \mathbf{D}_{t,a}^1 + \mathbf{D}_{t,a}'' \right) \right\|_2$$

$$\leq \sum_{l=a}^{L-1} \underbrace{\left\| \mathbf{W}_L^1 \mathbf{D}_{t,L-1}^1 \cdots \mathbf{W}_{l+1}^1 \mathbf{D}_{t,l}^{0/1} \right\|_2}_{Allen-Zhu\,et\,al.\,(2019, Lemma\,7.4(a))} \cdot \left\| \mathbf{D}_{t,l}'' \right\|_2 \cdot \underbrace{\left\| \mathbf{D}_{t,l}^{0/1} \left( \mathbf{W}_l^1 + \mathbf{W}_l'' \right) \cdots \left( \mathbf{D}_{t,a}^1 + \mathbf{D}_{t,a}'' \right) \right\|_2}_{Allen-Zhu\,et\,al.\,(2019, Lemma\,8.6(b))}$$

$$+ \sum_{l=a+1}^{L} \underbrace{\left\| \mathbf{W}_{L-1}^1 \cdots \mathbf{W}_{l+1}^1 \mathbf{D}_{t,l}^1 \right\|_2}_{Allen-Zhu\,et\,al.\,(2019, Lemma\,7.4(b))} \cdot \left\| \mathbf{W}_l'' \right\|_2 \cdot \underbrace{\left\| \left( \mathbf{D}_{t,l-1}^1 + \mathbf{D}_{t,l-1}'' \right) \left( \mathbf{W}_{l-1}^1 + \mathbf{W}_{l-1}'' \right) \cdots \left( \mathbf{D}_{t,a}^1 + \mathbf{D}_{t,a}'' \right) \right\|_2}_{Allen-Zhu\,et\,al.\,(2019, Lemma\,8.6(b))}$$

$$\leq L \cdot \mathcal{O}\left( \sqrt{\frac{m\omega^{2/3}L\log(m)}{K}} \right) \cdot \mathcal{O}(\sqrt{L}) + L \cdot \mathcal{O}\left( \sqrt{\frac{m}{K}} \right) \cdot \omega \cdot \mathcal{O}(\sqrt{L})$$

$$= \mathcal{O}\left( \frac{\omega^{1/3} L^2 \sqrt{m\log(m)}}{\sqrt{K}} \right) + \mathcal{O}\left( \omega L^{3/2} \sqrt{\frac{m}{K}} \right) = \mathcal{O}\left( \frac{\omega^{1/3} L^2 \sqrt{m\log(m)}}{\sqrt{K}} \right),$$

where the structure of $\mathbf{D}_{t,l}^{0/1}$ is same as the one in Allen-Zhu et al. (2019). We direct the reader to the corresponding literature for any interest. $\qquad \square$

**Lemma 8.** *(Cao & Gu, 2019, Lemma D.1) If $\omega \leq \mathcal{O}(L^{-9/2}[\log(m)]^{-3/2})$, then with probability at least $1 - \mathcal{O}(TL) \cdot \exp\left(-\Omega(m\omega^{2/3}L)\right)$ over $\mathcal{W}^1$, for all $\mathcal{W}, \widetilde{\mathcal{W}} \in \mathcal{B}(\mathcal{W}^1, \omega), t \in [T]$ and $l \in [L-1]$,*

$$\|\mathbf{D}_{t,l} - \widetilde{\mathbf{D}}_{t,l}\|_0 \leq \mathcal{O}(m\omega^{2/3}L).$$

We will use Lemmas 7 and 8 to prove the below Theorem 9.

**Theorem 9.** *Let $\mathcal{B}(\mathcal{W}^1, \omega) := \{\mathcal{W} : \|\mathbf{W}_l - \mathbf{W}_l^1\|_2 \leq \omega, l \in [L]\}$ be an $\omega$-neighborhood of the initialization. There exists an absolute constant $\kappa$ such that, with probability at least $1 - \mathcal{O}(TL^2)\exp(-\Omega(m\omega^{2/3}L))$ over the randomness of $\mathcal{W}^1$, for all $t \in [T]$, and $\mathcal{W}, \mathcal{W}' \in \mathcal{B}(\mathcal{W}^1, \omega)$ with $\omega \leq \kappa L^{-6}[\log(m)]^{-3/2}$, it holds uniformly that*

$$\left\| \boldsymbol{f}_{\mathcal{W}}(\boldsymbol{X}_t) - \boldsymbol{f}_{\widetilde{\mathcal{W}}}(\boldsymbol{X}_t) - \langle \nabla_{\widetilde{\mathcal{W}}} \boldsymbol{f}_{\widetilde{\mathcal{W}}}(\boldsymbol{X}_t), \mathcal{W} - \widetilde{\mathcal{W}} \rangle \right\|_2 \leq \mathcal{O}\left( \frac{\omega^{4/3}L^3\sqrt{m\log(m)}}{\sqrt{K}} \right).$$

*Proof of Theorem 9:* For $\mathcal{W}, \widetilde{\mathcal{W}} \in \mathcal{B}(\mathcal{W}^1, \omega) := \{\mathcal{W} : \|\mathbf{W}_l - \mathbf{W}_l^1\|_2 \leq \omega, l \in [L]\}$, we have

$$\boldsymbol{f}_{\mathcal{W}}(\boldsymbol{x}_t) - \boldsymbol{f}_{\widetilde{\mathcal{W}}}(\boldsymbol{x}_t) - \langle \nabla_{\widetilde{\mathcal{W}}} \boldsymbol{f}_{\widetilde{\mathcal{W}}}(\boldsymbol{x}_t), \mathcal{W} - \widetilde{\mathcal{W}} \rangle$$

$$= \mathbf{W}_L \boldsymbol{h}_{t,L-1} - \widetilde{\mathbf{W}}_L \tilde{\boldsymbol{h}}_{t,L-1} - \langle \tilde{\boldsymbol{h}}_{t,L-1}, \mathbf{W}_L - \widetilde{\mathbf{W}}_L \rangle - \sum_{l=1}^{L-1} \widetilde{\mathbf{W}}_L \left( \prod_{r=l+1}^{L-1} \widetilde{\mathbf{D}}_{t,r}\widetilde{\mathbf{W}}_r \right) \widetilde{\mathbf{D}}_{t,l}(\mathbf{W}_{l-1} - \widetilde{\mathbf{W}}_{l-1})\tilde{\boldsymbol{h}}_{t,l-1}$$

$$= \mathbf{W}_L(\boldsymbol{h}_{t,L-1} - \tilde{\boldsymbol{h}}_{t,L-1}) - \sum_{l=1}^{L-1} \widetilde{\mathbf{W}}_L \left( \prod_{r=l+1}^{L-1} \widetilde{\mathbf{D}}_{t,r}\widetilde{\mathbf{W}}_r \right) \widetilde{\mathbf{D}}_{t,l}(\mathbf{W}_{l-1} - \widetilde{\mathbf{W}}_{l-1})\tilde{\boldsymbol{h}}_{t,l-1}.$$

Similar to Claim 11.2 in Allen-Zhu et al. (2019), there exists $\mathbf{D}'_{i,l}$ such that

$$\tilde{\boldsymbol{h}}_{t,L-1} - \boldsymbol{h}_{t,L-1} = \sum_{l=1}^{L-1} \left( \prod_{r=l+1}^{L-1} (\mathbf{D}_{t,r} + \mathbf{D}'_{t,r})\mathbf{W}_r \right) (\mathbf{D}_{t,l} + \mathbf{D}'_{t,l})(\widetilde{\mathbf{W}}_{l-1} - \mathbf{W}_{l-1})\tilde{\boldsymbol{h}}_{t,l-1},$$

where diagonal matrices $\mathbf{D}'_{t,l}, l \in [L-1]$ has entries $[-1, 1]$ such that $\|\mathbf{D}'_{t,l}\|_0 \leq \mathcal{O}(m\omega^{2/3}L)$. Therefore,

$$\left\| \boldsymbol{f}_{\mathcal{W}}(\boldsymbol{x}_t) - \boldsymbol{f}_{\widetilde{\mathcal{W}}}(\boldsymbol{x}_t) - \langle \nabla_{\widetilde{\mathcal{W}}} \boldsymbol{f}_{\widetilde{\mathcal{W}}}(\boldsymbol{x}_t), \mathcal{W} - \widetilde{\mathcal{W}} \rangle \right\|_2$$

$$= \left\| \sum_{l=1}^{L-1} \left( \prod_{r=l+1}^{L} \mathbf{W}_r(\mathbf{D}_{t,r-1} + \mathbf{D}'_{t,r-1}) \right)(\mathbf{W}_{l-1} - \widetilde{\mathbf{W}}_{l-1})\tilde{\boldsymbol{h}}_{t,l-1} - \sum_{l=1}^{L-1} \left( \prod_{r=l+1}^{L} \widetilde{\mathbf{W}}_r \widetilde{\mathbf{D}}_{t,r-1} \right)(\mathbf{W}_{l-1} - \widetilde{\mathbf{W}}_{l-1})\tilde{\boldsymbol{h}}_{t,l-1} \right\|_2$$

$$\leq \max_r \left\| \prod_{r=l+1}^{L} \mathbf{W}_r(\mathbf{D}_{t,r-1} + \mathbf{D}'_{t,r-1}) - \prod_{r=l+1}^{L} \widetilde{\mathbf{W}}_r \widetilde{\mathbf{D}}_{t,r-1} \right\|_2 \cdot \sum_{l=1}^{L-1} \|\mathbf{W}_{l-1} - \widetilde{\mathbf{W}}_{l-1}\|_2 \cdot \|\tilde{\boldsymbol{h}}_{t,l-1}\|_2$$

$$\overset{\textcircled{1}}{\leq} \mathcal{O}\left( \frac{\omega^{1/3}L^2\sqrt{m\log(m)}}{\sqrt{K}} \right) \cdot \omega L = \mathcal{O}\left( \frac{\omega^{4/3}L^3\sqrt{m\log(m)}}{\sqrt{K}} \right),$$

where $\textcircled{1}$ is due to

$$\left\| \prod_{r=l+1}^{L} \mathbf{W}_r(\mathbf{D}_{t,r-1} + \mathbf{D}'_{t,r-1}) - \prod_{r=l+1}^{L} \widetilde{\mathbf{W}}_r \widetilde{\mathbf{D}}_{t,r-1} \right\|_2$$

$$\leq \left\| \prod_{r=l+1}^{L} \mathbf{W}_r(\mathbf{D}_{t,r-1} + \mathbf{D}'_{t,r-1}) - \prod_{r=l+1}^{L} \mathbf{W}_r^1 \mathbf{D}_{t,r-1}^1 \right\|_2 + \left\| \prod_{r=l+1}^{L} \widetilde{\mathbf{W}}_r \widetilde{\mathbf{D}}_{t,r-1} - \prod_{r=l+1}^{L} \mathbf{W}_r^1 \mathbf{D}_{t,r-1}^1 \right\|_2$$

$$\overset{\textcircled{2}}{\leq} \mathcal{O}\left( \frac{\omega^{1/3}L^2\sqrt{m\log(m)}}{\sqrt{K}} \right),$$

where ② holds by using Lemma 7. This is because $\widetilde{\mathcal{W}}, \mathcal{W} \in \mathcal{B}(\mathcal{W}^1, \omega)$. By Lemma 8, we have $\|\widetilde{\mathbf{D}}_{t,r-1} - \mathbf{D}^1_{t,r-1}\|_0 \leq \mathcal{O}(m\omega^{2/3}L)$ and $\|\mathbf{D}_{t,r-1} - \mathbf{D}^1_{t,r-1}\|_0 \leq \mathcal{O}(m\omega^{2/3}L)$. Hence $\|\mathbf{D}_{t,r-1} + \mathbf{D}'_{t,r-1} - \mathbf{D}^1_{t,r-1}\|_0 \leq \|\mathbf{D}_{t,r-1} - \mathbf{D}^1_{t,r-1}\|_0 + \|\mathbf{D}'_{t,r-1}\|_0 \leq \mathcal{O}(m\omega^{2/3}L)$ for any $r \in [L-1]$. $\qquad\square$

The below Proposition 10 is directly derived based on Theorem 9 by utilizing the convexity and Lipchitzness of a given function $\ell$. This proposition implies $\ell$ is near convex with respect to the neural network parameter $\mathcal{W}$, although the neural network is not convex.

**Proposition 10.** *Let the $\ell : \mathbb{R}^K \to \mathbb{R}$ be a convex and c-Lipschitz function. There exists an absolute constant $\kappa$ such that, with probability at least $1 - \mathcal{O}(TL^2)\exp(-\Omega(m\omega^{2/3}L))$ over the randomness of $\mathcal{W}^1$, for all $t \in [T]$, and $\mathcal{W}, \mathcal{W}' \in \mathcal{B}(\mathcal{W}^1, \omega)$ with $\omega \leq \kappa L^{-6}[\log(m)]^{-3/2}$, it holds uniformly that*

$$\ell(\boldsymbol{f}_{\mathcal{W}'}(\boldsymbol{X}_t)) - \ell(\boldsymbol{f}_{\mathcal{W}}(\boldsymbol{X}_t)) \leq \langle \nabla_{\mathcal{W}}\ell(\boldsymbol{f}_{\mathcal{W}'}(\boldsymbol{X}_t)), \mathcal{W}' - \mathcal{W}\rangle + \mathcal{O}\left(\frac{c\omega^{4/3}L^3\sqrt{m\log(m)}}{\sqrt{K}}\right).$$

*Proof of Proposition 10:*

$$
\begin{aligned}
&\ell(\boldsymbol{f}_{\mathcal{W}'}(\boldsymbol{X}_t)) - \ell(\boldsymbol{f}_{\mathcal{W}}(\boldsymbol{X}_t)) \\
&\leq -\ell'(\boldsymbol{f}_{\mathcal{W}'}(\boldsymbol{X}_t))^\top (\boldsymbol{f}_{\mathcal{W}}(\boldsymbol{X}_t) - \boldsymbol{f}_{\mathcal{W}'}(\boldsymbol{X}_t)) \qquad \text{b/c } \ell \text{ is convex} \\
&= -\ell'(\boldsymbol{f}_{\mathcal{W}'}(\boldsymbol{X}_t))^\top (\boldsymbol{f}_{\mathcal{W}}(\boldsymbol{X}_t) - \boldsymbol{f}_{\mathcal{W}'}(\boldsymbol{X}_t) - \langle \nabla_{\mathcal{W}}\boldsymbol{f}_{\mathcal{W}'}(\boldsymbol{X}_t), \mathcal{W} - \mathcal{W}'\rangle) \\
&\quad + \ell'(\boldsymbol{f}_{\mathcal{W}'}(\boldsymbol{X}_t))^\top \langle \nabla_{\mathcal{W}}\boldsymbol{f}_{\mathcal{W}'}(\boldsymbol{X}_t), \mathcal{W}' - \mathcal{W}\rangle \\
&\leq \mathcal{O}\left(\frac{c\omega^{4/3}L^3\sqrt{m\log(m)}}{\sqrt{K}}\right) + \langle \nabla_{\mathcal{W}}\ell(\boldsymbol{f}_{\mathcal{W}'}(\boldsymbol{X}_t)), \mathcal{W}' - \mathcal{W}\rangle,
\end{aligned}
\tag{14}
$$

where the last inequality holds due to the Lipschitzness of $\ell(\cdot)$ and the usage of Theorem 9. $\qquad\square$

The below Theorem 11 shows the boundness for the gradient of the neural network with respect to the parameters $\mathcal{W}$.

**Theorem 11.** *There exists an absolute constant $\kappa$ such that, with probability at least $1 - \mathcal{O}(TL^2)\exp(-\Omega(m\omega^{2/3}L))$ over the randomness of $\mathcal{W}^1$, for all $t \in [T], l \in [L]$, and $\mathcal{W} \in \mathcal{B}(\mathcal{W}^1, \omega)$ with $\omega \leq \kappa L^{-6}[\log(m)]^{-3}$, it holds uniformly that $\|\nabla_{\mathcal{W}}\boldsymbol{f}_{\mathcal{W}}(\boldsymbol{X}_t)\|_2 = \mathcal{O}\big(K^{1/2}\omega^{1/3}L^{5/2}\sqrt{m\log(m)}\big) \leq \mathcal{O}\big(\sqrt{LKm}\big)$.*

*Proof of Theorem 11:* This theorem is adapted from Cao & Gu (2019, Lemma B.3). From Equation (13) we have

$$\|\nabla_{\mathbf{W}_L}\boldsymbol{f}_{\mathcal{W}}(\boldsymbol{x}_t)\|_2^2 = \sum_{k=1}^K \|\nabla_{\mathbf{W}_L}f_{\mathcal{W}}^k(\boldsymbol{x}_t)\|_2^2 = \sum_{k=1}^K \|\boldsymbol{e}_k\boldsymbol{h}_{t,L-1}^\top\|_2^2 = \mathcal{O}(K),$$

and for $l \in [L-1]$,

$$
\begin{aligned}
\|\nabla_{\mathbf{W}_l} \boldsymbol{f}_{\mathcal{W}}(\boldsymbol{x}_t)\|_2 &= \sum_{k=1}^{K} \|\boldsymbol{h}_{t,l-1} \mathbf{e}_k^\top \mathbf{W}_L \Big(\prod_{r=l+1}^{L-1} \mathbf{D}_{t,r} \mathbf{W}_r\Big) \mathbf{D}_{t,l}\|_2 \\
&\leq \sum_{k=1}^{K} \|\boldsymbol{h}_{t,l-1}\|_2 \cdot \|\mathbf{e}_k^\top \mathbf{W}_L \Big(\prod_{r=l+1}^{L-1} \mathbf{D}_{t,r} \mathbf{W}_r\Big) \mathbf{D}_{t,l}\|_2 \\
&\leq \sum_{k=1}^{K} \mathcal{O}(1) \cdot \|\prod_{r=l+1}^{L} \mathbf{W}_r \mathbf{D}_{t,r-1}\|_2 \\
&\leq \mathcal{O}(K) \cdot \|\prod_{r=l+1}^{L} \mathbf{W}_r \mathbf{D}_{t,r-1}\|_2 \\
&\leq \mathcal{O}(K) \cdot \Big[\|\prod_{r=l+1}^{L} \mathbf{W}_r \mathbf{D}_{t,r-1} - \prod_{r=l+1}^{L} \mathbf{W}_r^1 \mathbf{D}_{t,r-1}^1\|_2 + \|\prod_{r=l+1}^{L} \mathbf{W}_r^1 \mathbf{D}_{t,r-1}^1\|_2\Big] \\
&\leq \mathcal{O}(K) \cdot \Big[\mathcal{O}\Big(\frac{\omega^{1/3} L^2 \sqrt{m \log(m)}}{\sqrt{K}}\Big) + \mathcal{O}\Big(\sqrt{\frac{m \omega^{2/3} L \log(m)}{K}}\Big)\Big] \\
&= \mathcal{O}\Big(K^{1/2} \omega^{1/3} L^2 \sqrt{m \log(m)}\Big) \leq \mathcal{O}(\sqrt{Km}),
\end{aligned}
\tag{15}
$$

where (15) holds due to Lemma 7 and Lemma 7.4(a) in Allen-Zhu et al. (2019), and the last equality holds due to the upper bound of $\omega$. Consequently,

$$
\|\nabla_{\mathcal{W}} \boldsymbol{f}_{\mathcal{W}}(\boldsymbol{X}_t)\|_2 = \mathcal{O}\Big(K^{1/2} \omega^{1/3} L^{5/2} \sqrt{m \log(m)}\Big) \leq \mathcal{O}\Big(\sqrt{LKm}\Big).
\tag{16}
$$

$\square$

*Proof of Theorem 6:* Note that the hinge loss $\ell_1$ is a convex 1-Lipschitz function. It is clear to see that $\ell(\boldsymbol{u})$ is convex with respect to $\boldsymbol{u} \in \mathbb{R}^K$. To show its Lipschitzness, take $\boldsymbol{u}, \tilde{\boldsymbol{u}} \in \mathbb{R}^K$. Thus,

$$
\begin{aligned}
&|\ell(\boldsymbol{u}) - \ell(\tilde{\boldsymbol{u}})| \\
&= \Big|\frac{\lambda}{K} \sum_{k=1}^{K} (\ell_1(u_k) - \ell_1(\tilde{u}_k)) + (1-\lambda)(\ell_1(\max_{k \in [K]} u_k) - \ell_1(\max_{k \in [K]} \tilde{u}_k))\Big| \\
&\leq \frac{\lambda}{K} \Big|\sum_{k=1}^{K} (\ell_1(u_k) - \ell_1(\tilde{u}_k))\Big| + (1-\lambda)\Big|\max_{k \in [K]} (u_k - \tilde{u}_k)\Big| \\
&\leq \frac{\lambda}{\sqrt{K}} \|\boldsymbol{u} - \tilde{\boldsymbol{u}}\|_2 + (1-\lambda)\|\boldsymbol{u} - \tilde{\boldsymbol{u}}\|_2 \\
&= (\frac{\lambda}{\sqrt{K}} + 1 - \lambda)\|\boldsymbol{u} - \tilde{\boldsymbol{u}}\|_2
\end{aligned}
$$

implies $\ell(\cdot)$ is a function with Lipschitz constant $\frac{\lambda}{\sqrt{K}} + 1 - \lambda$, and hence $\|\ell'(\boldsymbol{u})\|_2 \leq \frac{\lambda}{\sqrt{K}} + 1 - \lambda$. Additionally, assume $\lambda_{t,k} \leq C_\lambda, \Delta_{t,k} \leq C_\Delta$ for all $t \in [T], k \in [K]$, thus $\mathcal{L}(\boldsymbol{X}; \mathcal{W})$ is Lipschitz mapping with respect to $\boldsymbol{f}_{\mathcal{W}}(\boldsymbol{X})$ with Lipschitz constant $\frac{\lambda}{\sqrt{K}} + 1 - \lambda + \sqrt{K} C_\lambda C_\Delta$.

Recall the risk function then can be rewritten as

$$
\ell(\boldsymbol{f}_{\mathcal{W}^t}(\boldsymbol{X}_t)) + \sum_{k=1}^{K} \lambda_{t,k} \cdot \Delta_{t,k} \ell_{2,\gamma}(f_{\mathcal{W}^t}^k(\boldsymbol{X}_t)),
$$

and the updating rule is

$$\mathcal{W}^{t+1} = \mathcal{W}^t - \eta_1 \left[ \nabla_{\mathcal{W}} \ell(\boldsymbol{f}_{\mathcal{W}^t}(\boldsymbol{X}_t)) + \sum_{k=1}^{K} \lambda_{t,k} \Delta_{t,k} \nabla_{\mathcal{W}} \ell_{2,\gamma}(f_{\mathcal{W}^t}^k(\boldsymbol{x}_t)) \right].$$

For notation simplicity, substitute $\omega$ in the first term of (14) as $\frac{R}{\sqrt{m}}$ and let

$$\nu = \mathcal{O}\left( \frac{(\lambda/\sqrt{K} + 1 - \lambda)R^{4/3}L^3\sqrt{\log(m)}}{m^{1/6}\sqrt{K}} \right).$$

Therefore, for $\mathcal{W}^* \in \mathcal{B}(\mathcal{W}^1, \frac{R}{\sqrt{m}})$, we have $\|\mathcal{W}^* - \mathcal{W}^1\|_2 \le \frac{R\sqrt{L}}{\sqrt{m}}$, and

$$\frac{1}{T}\sum_{t=1}^{T}\ell(\boldsymbol{f}_{\mathcal{W}^t}(\boldsymbol{x}_t)) - \frac{1}{T}\sum_{t=1}^{T}\ell(\boldsymbol{f}_{\mathcal{W}^*}(\boldsymbol{x}_t))$$

$$\le \frac{1}{T}\sum_{t=1}^{T}\langle\nabla_{\mathcal{W}}\ell(\boldsymbol{f}_{\mathcal{W}^t}(\boldsymbol{x}_t)), \mathcal{W}^t - \mathcal{W}^*\rangle + \nu \qquad \text{b/c of Proposition 10}$$

$$\le \frac{1}{\eta_1 T}\sum_{t=1}^{T}\langle\mathcal{W}^t - \mathcal{W}^{t+1}, \mathcal{W}^t - \mathcal{W}^*\rangle + \frac{1}{T}\sum_{t=1}^{T}\sum_{k=1}^{K}\langle\lambda_{t,k}\Delta_{t,k}\nabla_{\mathcal{W}}\ell_{2,\gamma}(f_{\mathcal{W}^t}^k(\boldsymbol{x}_t)), \mathcal{W}^* - \mathcal{W}^t\rangle + \nu$$

$$\le \frac{1}{2\eta_1 T}\sum_{t=1}^{T}(\|\mathcal{W}^t - \mathcal{W}^{t+1}\|_2^2 + \|\mathcal{W}^t - \mathcal{W}^*\|_2^2 - \|\mathcal{W}^{t+1} - \mathcal{W}^*\|_2^2)$$

$$+ \frac{1}{T}\sum_{t=1}^{T}\sum_{k=1}^{K}\langle\lambda_{t,k}\Delta_{t,k}\nabla_{\mathcal{W}}\ell_{2,\gamma}(f_{\mathcal{W}^t}^k(\boldsymbol{x}_t)), \mathcal{W}^* - \mathcal{W}^t\rangle + \nu$$

$$\le \frac{\eta_1^2}{2\eta_1}(\frac{\lambda}{\sqrt{K}} + 1 - \lambda + \sqrt{K}C_\lambda C_\Delta)^2 \cdot \mathcal{O}(KLm) + \frac{\omega^2 L}{2\eta_1 T}$$

$$+ \frac{1}{T}\sum_{t=1}^{T}C_\lambda C_\Delta \cdot |\ell_{2,\gamma}'(\cdot)|\sqrt{K} \cdot \left\|\nabla_{\mathcal{W}^t}\boldsymbol{f}_{\mathcal{W}^t}(\boldsymbol{X}_t)\right\|_2 \cdot \sqrt{L}\omega + \nu,$$

$$\le \frac{\eta_1}{2}(\frac{\lambda}{\sqrt{K}} + 1 - \lambda + \sqrt{K}C_\lambda C_\Delta)^2 \cdot \mathcal{O}(KLm) + \frac{\omega^2 L}{2\eta_1 T}$$

$$+ C_\lambda C_\Delta \cdot \mathcal{O}\left(\sqrt{KL}\omega \cdot K^{1/2}\omega^{1/3}L^{5/2}\sqrt{m\log(m)}\right) + \nu,$$

$$\le (\frac{\lambda}{\sqrt{K}} + 1 - \lambda + \sqrt{K}C_\lambda C_\Delta)^2 \cdot \mathcal{O}\left(\frac{\eta_1}{2}KLm\right) + \frac{LR^2}{2\eta_1 Tm}$$

$$+ C_\lambda C_\Delta \cdot \mathcal{O}\left(\frac{KL^3 R^{4/3}\sqrt{\log(m)}}{m^{1/6}}\right) + \mathcal{O}\left(\frac{(\lambda/\sqrt{K} + 1 - \lambda)R^{4/3}L^3\sqrt{\log(m)}}{m^{1/6}\sqrt{K}}\right),$$

where the last inequality is due to telescoping sum, Proposition 10, and the below fact

$$\|\mathcal{W}^t - \mathcal{W}^{t+1}\|_2^2 = \eta_1^2\left\|\nabla_{\mathcal{W}}\mathcal{L}(\boldsymbol{X}_t, \mathcal{W}^t)\right\|_2^2$$

$$\le \eta_1^2\left\|\nabla_{\boldsymbol{f}_{\mathcal{W}^t}(\boldsymbol{X}_t)}\mathcal{L}(\boldsymbol{X}_t, \mathcal{W}^t)\right\|_2^2 \cdot \left\|\nabla_{\mathcal{W}^t}\boldsymbol{f}_{\mathcal{W}^t}(\boldsymbol{X}_t)\right\|_2^2$$

$$\le \eta_1^2 \cdot (\frac{\lambda}{\sqrt{K}} + 1 - \lambda + \sqrt{K}C_\lambda C_\Delta)^2 \cdot \mathcal{O}\left(K\omega^{2/3}L^5 m\log(m)\right), \tag{17}$$

where (17) holds due to the Lipschitzness of the function $\mathcal{L}(\boldsymbol{X}_t; \mathcal{W}^t)$ with respect to $\boldsymbol{f}_{\mathcal{W}^t}(\boldsymbol{X}_t)$ and the usage of (16).

Thus, by setting a constant $c^* := (\frac{\lambda}{\sqrt{K}} + 1 - \lambda + \sqrt{K}C_\lambda C_\Delta)^2 + \frac{\lambda}{K} + \frac{1-\lambda}{\sqrt{K}} + KC_\lambda C_\Delta$ and choosing $\eta_1 = \frac{R}{\sqrt{K}Tm}$, we have

$$\frac{1}{T}\sum_{t=1}^{T}\ell(\boldsymbol{f}_{\mathcal{W}^t}(\boldsymbol{X}_t)) - \frac{1}{T}\sum_{t=1}^{T}\ell(\boldsymbol{f}_{\mathcal{W}^*}(\boldsymbol{X}_t)) \le c^*\mathcal{O}\left(\frac{LR\sqrt{K}}{\sqrt{T}} + \frac{L^3R^{4/3}\sqrt{\log(m)}}{m^{1/6}}\right).$$

$\square$

