# OpenReview forum: "Generalized Prediction Set with Bandit Feedback"
_TMLR — Accepted by TMLR_

### Review · Reviewer_WqAu · 2025-02-09

**Summary Of Contributions:**

This paper extends the generalized prediction set (GPS) problem to the online setting with bandit feedback. The authors propose an algorithm for this new setting and analyze its properties under certain assumptions. Experimental results show that the proposed algorithms outperform baselines.

**Audience:**

Yes

**Claims And Evidence:**

Yes

**Requested Changes:**

Please see my comments above.

**Strengths And Weaknesses:**

Strengths: It nice to see that this paper extends the generalized prediction set problem to the online setting with bandit feedback. The authors also conducted several experiments to empirically verify the effectiveness of the proposed algorithm.

Weaknesses and Questions:
1. I'm not sure if I fully understand why $\lambda_{t,k}$ should change over time. It seems that risk function (Eq. (7)) you try to optimize is changing with respect to your algorithm (Algorithm 1), which is weird.
2. Assumption 3 seems strong to me: both $\lambda_{t,k}$ and $\Delta_{t,k}$ are changing with respect to the proposed algorithm; their upper bounds should be proven, instead of assumed.
3. It seems to me that many theoretical results can be vacuous: e.g., in Theorem 5, the value of $p_k$ can be very small which makes the recall meaningless. Similarly, it's not clear how large is the value of $c^*$ in Theorem 6 since it depends on $C_\lambda$ and $C_{\Delta}$.

---

> ### Author Response · Authors · 2025-02-26
>
> > I'm not sure if I fully understand why $\lambda_{t,k}$ should change over time. It seems that risk function (Eq. (7)) you try to optimize is changing with respect to your algorithm (Algorithm 1), which is weird.
>
> We appreciate the reviewer’s thoughtful question and apologize for any lack of clarity in our original manuscript. The dynamic adjustment of  $\lambda_{t,k}$  is necessary to ensure that the recall for normal classes remains above  $1 - \gamma$. A fixed  $\lambda_{t,k}$  presents several challenges: i) If set too small, the recall constraint for normal classes may not be satisfied. ii) If set too large, the recall is overly conservative, leading to excessively large prediction sets, which is undesirable. iii) The appropriate value of  $\lambda_{t,k}$  can vary across tasks, making it difficult to predefine an optimal value.
>
> To address these challenges, we adopt a primal-dual optimization approach, allowing  $\lambda_{t,k}$  to dynamically adjust during learning. This ensures that $ \lambda_{t,k}$  neither diminishes to an ineffective level nor becomes excessively large. Similar methods have been successfully applied in constrained learning settings, as discussed in the following references [1, 2]. We have incorporated additional clarification in our revision to better explain this aspect of the algorithm.
>
>
> [1] Yu, Hao, Michael Neely, and Xiaohan Wei. "Online convex optimization with stochastic constraints." Advances in Neural Information Processing Systems 30 (2017).
>
> [2] Chamon, Luiz FO, et al. "Constrained learning with non-convex losses." IEEE Transactions on Information Theory 69.3 (2022): 1739-1760.
>
>
> > Assumption 3 seems strong to me: both $\lambda_{t,k}$ and $\Delta_{t,k}$ are changing with respect to the proposed algorithm; their upper bounds should be proven, instead of assumed.
>
> We appreciate the reviewer’s valuable insights regarding Assumption 3. We acknowledge that the assumption on the boundedness of  $\lambda_{t,k}$  may be strong, and proving it rigorously is an important direction for future work. We will investigate and establish its theoretical upper bound in our next research project.
>
> Regarding  $\Delta_{t,k}$, we believe that its boundedness assumption is mild. Specifically, $\Delta_{t, k}:=\frac{\Bbb{1}\\{A_t=k\\}}{\pi_t(k\mid \boldsymbol X_t)}\Bbb{1}\\{A_t=Y_t\\}$ is primarily influenced by the policy  $\pi_t(k | \boldsymbol X_t)$, which appears in the denominator. Since $\pi_t$  is user-defined, practitioners can ensure that  $\pi_t(k | \boldsymbol X_t)$  does not approach zero by setting a minimum probability threshold. This practical constraint mitigates concerns regarding the boundedness of  $\Delta_{t,k}$. For example, if the minimum value of the policy is set as $\underline{p}$, then $\Delta_{t,k}\leq \max\\{\frac{1}{\underline{p}},\frac{1}{1-K\underline{p}}\\}$.
>
> > It seems to me that many theoretical results can be vacuous: e.g., in Theorem 5, the value of $p_k$ can be very small which makes the recall meaningless. Similarly, it's not clear how large is the value of $c^*$ in Theorem 6 since it depends on $C_\lambda$ and $C_\Delta$.
>
> We appreciate the reviewer’s insightful observation. Intuitively, if  $p_k$ is very small, it means that we may not collect sufficient samples for class $k$  within a finite time horizon, leading to potential failures in recall control. However, the denominator in Theorem 5 includes $T$,  compensating for this issue in the long run. Nevertheless, we acknowledge that in a short-term regime, the recall bound may be less meaningful. To clarify this limitation, we have explicitly discussed this scenario in the revised manuscript.
>
> Regarding the value of  $c^*$  in Theorem 6, we recognize that its dependence on constant $C_\lambda$ may raise concerns. While bounding this term is non-trivial (we may add additional assumptions), we plan to include detailed proof in future work to establish a clearer understanding of their behavior.
>
>
> > Requested Changes:  Please see my comments above.
>
> We sincerely appreciate the reviewer’s constructive feedback. Based on your comments, we have added further clarifications in the revised manuscript (see colored text on Pages 6 and 7) to address potential concerns and improve readability. Specifically, we have: i) Expanded the explanation of why $\lambda_{t,k}$  changes over time to highlight its role in recall control and the necessity of dynamic adjustment. ii) Clarified the boundedness of  $\Delta_{t,k}$  and why it remains well-behaved in practical settings. iii) Discussed the implications of small $p_k$  values in Theorem 5 and acknowledged the short-term limitations.

---

### Review · Reviewer_f3Wq · 2025-02-17

**Summary Of Contributions:**

The authors consider the problem of set-valued prediction that also allows for OOD under bandit feedback in multi-class classification settings. In particular, they propose training a neural network with a loss function aiming to satisfy each of the following objectives: minimizing prediction set size, guaranteeing class-conditional coverage, and detecting  as many OOD samples as possible. The authors use a threshold rule on the output of such a neural network to construct prediction sets satisfying the above objectives. They further show theoretically  that their algorithm achieves asymptotically class-conditional coverage as well as they provide a regret upper bound for their algorithm.

**Audience:**

Yes

**Broader Impact Concerns:**

No concerns.

**Claims And Evidence:**

Yes

**Requested Changes:**

As the authors do have quite some space (the paper is less than 10 pages), they could bring in the main the additional experimental results on the hold-out set. In addition, they could provide the results for the different metrics in separate figures, so that the comparison can be easier for the reader.

**Strengths And Weaknesses:**

**Strenghts.** The proposed method appears quite interesting and seems to tackle an important problem, extending prediction sets that also account for OOD detection. The paper appears well written. It is appropriately placed among contemporary literature, it is well motivated and clearly organized. The presentation of the contributions and experiments is quite clear. The experimental evaluation appears comprehensive and the results convincing. The authors conducted experiments using multiple datasets, comparing with competitive baselines while using many different metrics. They also conducted ablation on the choice of the lambda parameter controlling the trade-off between performance in OOD and the informativeness (size) of the prediction sets.

**Weaknesses.** The recall results for accum_normal_min and accum_nomral_max look a bit cluttered and it makes it a bit difficult for the reader to compare. The results on the hold-out set that seem important as one could view this as the test time performance of the proposed method are in the appendix.

---

> ### Author Response · Authors · 2025-02-26
>
> > The recall results for accum\_normal\_min and accum\_nomral\_max look a bit cluttered and it makes it a bit difficult for the reader to compare. The results on the hold-out set that seem important as one could view this as the test time performance of the proposed method are in the appendix.
>
> We sincerely appreciate your feedback on the clarity of our experimental presentation. To improve readability and facilitate comparison, we have separately plotted recall curves for normal\_min and normal\_max in the comparison and ablation study, covering both the training and holdout set. Additionally, to highlight the importance of test-time performance, we have relocated Figures 4 and 6 (holdout dataset performance) from the appendix to the main article.
>
> > As the authors do have quite some space (the paper is less than 10 pages), they could bring in the main the additional experimental results on the hold-out set. In addition, they could provide the results for the different metrics in separate figures, so that the comparison can be easier for the reader.
>
> We are grateful for your thoughtful suggestions on improving the clarity and accessibility of our results. In response, we have incorporated all figures from the appendix into the main article, ensuring that key results on the holdout set are more prominent. Furthermore, we have reorganized our figures by presenting different metrics in separate plots, making comparisons clearer and more intuitive for the reader.

---

### Review · Reviewer_uFKt · 2025-02-18

**Summary Of Contributions:**

This work examines set-valued prediction with out-of-distribution (OOD) detection under bandit feedback, where the agent can only observe the output for selected arms—meaning the label is only revealed when the current prediction is made. In this setting, the author proposes a novel algorithm that converges to the optimal policy at a rate of $1/\sqrt{T}$. Experimental results further demonstrate the efficiency of the proposed method.

**Audience:**

Yes

**Claims And Evidence:**

Yes

**Requested Changes:**

The current description of the bandit feedback setting is unclear and requires a more precise formulation. (See Weakness 1)

**Strengths And Weaknesses:**

Strengths:

1. The author first analyzes set-valued prediction with out-of-distribution (OOD) detection under bandit feedback and proposes an algorithm with theoretical guarantees.

2. Experimental results further validate the efficiency of the proposed method.

Weaknesses:

1. The bandit feedback setting lacks clarity. In traditional prediction, an action corresponds to making a prediction, and bandit feedback implies observing only whether the prediction is correct. However, in the context of set-valued prediction, it is unclear how actions should be defined for general sets. Should a label be randomly selected from the set, or should the entire set be treated as an action, close to combinatorial bandits? Moreover, the nature of the observations in this setting remains ambiguous. Additionally, when an empty set is predicted for an OOD case, does the action correspond to an empty set or a specific OOD label? Furthermore, can the bandit setting determine whether the OOD decision is correct?

2. In "Recall Control for Normal Classes," the author transforms the loss function using Bayes' theorem. However, this approach may not be valid when another loss function is considered simultaneously, as the interaction between multiple loss formulations is not well justified.

---

> ### Author Response · Authors · 2025-02-26
>
> > The bandit feedback setting lacks clarity. In traditional prediction, an action corresponds to making a prediction, and bandit feedback implies observing only whether the prediction is correct. However, in the context of set-valued prediction, it is unclear how actions should be defined for general sets. Should a label be randomly selected from the set, or should the entire set be treated as an action, close to combinatorial bandits? Moreover, the nature of the observations in this setting remains ambiguous. Additionally, when an empty set is predicted for an OOD case, does the action correspond to an empty set or a specific OOD label? Furthermore, can the bandit setting determine whether the OOD decision is correct?
>
> We sincerely appreciate the reviewer’s detailed questions and we apologize for having any less clarification for the bandit feedback setting.
>
>
> As for the question about actions defined in the context of set-valued prediction, in our framework, an action $A_t$ is always a single label selected from all possibilities (including $\text{OOD}$) using the policy defined below Eq. (6). Thus, our approach differs from combinatorial bandits, where an entire set would be treated as an action. That said, your question about combinatorial bandits makes sense and this is what we are working on for the subsequent research.
>
> In terms of the concern about the nature of the observations in this setting, besides feature information $\boldsymbol X_t$, the observation consists of bandit feedback, where we receive only binary correctness information for the selected action $A_t$, i.e., $\Bbb{1}\\{A_t=Y_t\\}$ in Eq. (5).
>
> For your question about "When an empty set is predicted for an OOD case, does the action correspond to an empty set or a specific OOD label", we want to clarify that if the model predicts an empty set in Step 2 (in Algorithm 1), it means the instance is suspected to be OOD. The empty set does not correspond to an action but rather acts as an intermediate uncertainty-aware prediction.
>
> The answer to the question "Can the bandit setting determine whether the OOD decision is correct" is Yes. If  $A_t = \text{OOD}$ and feedback is 1, the instance is confirmed as truly OOD.
>
> > In "Recall Control for Normal Classes," the author transforms the loss function using Bayes' theorem. However, this approach may not be valid when another loss function is considered simultaneously, as the interaction between multiple loss formulations is not well justified.
>
> We thank the reviewer’s insightful question. We understand the concern that introducing multiple loss terms (i.e., set size minimization, OOD detection, and recall control) could lead to unintended interactions. However, our formulation carefully balances these three losses. The interplay between these loss terms is controlled via the trade-off parameter $\lambda$ and the learning parameter $\lambda_{t,k}$ to control recalls in Eq. (7).
>
> To further validate the effectiveness of our recall control mechanism based on this transformation, we added Theorem 5 to show that, asymptotically, the recall remains above  $1 - \gamma$  as  $T \to \infty$. Additionally, Figures 3 and 4 provide empirical evidence that the recall constraints are well maintained across the training set and holdout set, Figures 5 and 6 show the desired normal recall even for different trade-off parameters $\lambda$.
>
> > The current description of the bandit feedback setting is unclear and requires a more precise formulation. (See Weakness 1)
>
> We appreciate the helpful suggestion of the reviewer to improve the readability of our article. We have added the clarification to the revision (see colored paragraph on Page 4).

---

### Decision · Action_Editor_ygBb · 2025-04-23

**Recommendation:** Accept with minor revision

**Comment:**

The paper studies set-valued prediction with out-of-distribution (OOD) detection under bandit feedback. The authors proposed BanditGPS algorithm which extends Generalized Prediction Set for online learning with OOD detection, analyzed its theoretical guarantee by proving a regret convergence rate of $O(T^{-1/2})$, and empirically validated its effectiveness.

The reviewers recognized the importance of problem, the interesting proposed algorithm, and the sound theoretical analysis. The reviewers also raised several concerns about unclear problem formulation, presentation issues, strong assumptions and confusing theoretical results. The authors addressed most of the concerns in response and revised paper, while one reviewer believes the concerns about strong Assumption 3 and explanation of Theorem 5 remain. Two reviewers are positive about the paper and one reviewer recommended rejection. Overall, AE shares the positive opinion about the paper while agreeing with reviewers ygBb's concern about assumption. Thus, the recommendation is 'Accept with minor revision'. The authors are suggested to further clarify in the camera-ready version about: 1) the limitation of Assumption 3 and potential theoretical solution of upper bounding $\lambda_{t,k}, \Delta_{t,k}$, and 2) relationship to combinatorial bandits and potential extension.

**Audience:**

Yes.

**Claims And Evidence:**

Yes, the claims are well supported by convincing evidence, including theorems and empirical validations.

---

> ### Author Response · Authors · 2025-05-04
> **Camera-Ready Submission Confirmation**
>
> Dear Action Editor,
>
> We sincerely thank you and the reviewers for your time, valuable feedback, and thoughtful coordination throughout the review process. We have submitted the camera-ready version of our paper and incorporated the suggested revisions in the newly added Section 6 (Discussion).
>
> Best regards,
> The Authors